# Dynamic Test-Time Compute Scaling in Control Policy: Difficulty-Aware Stochastic Interpolant Policy

**Inkook Chun**[†]    **Seungjae Lee**[‡]    **Michael S. Albergo**[◇]
**Saining Xie**[†]    **Eric Vanden-Eijnden**[†,§]
[†]New York University    [‡]University of Maryland
[◇]Harvard University    [§]Capital Fund Management

## Abstract

Diffusion- and flow-based policies deliver state-of-the-art performance on long-horizon robotic manipulation and imitation learning tasks. However, these controllers employ a **fixed inference budget at every control step**, regardless of task complexity, leading to computational inefficiency for simple subtasks while potentially underperforming on challenging ones. To address these issues, we introduce *Difficulty-Aware Stochastic Interpolant Policy (DA-SIP)*, a framework that enables robotic controllers to **adaptively adjust their integration horizon in real time** based on task difficulty. Our approach employs a *difficulty classifier* that analyzes RGB-D observations to dynamically select the step budget, the optimal solver variant, and ODE/SDE integration at each control cycle. DA-SIP builds upon the stochastic interpolant formulation to provide a unified framework that unlocks diverse training and inference configurations for diffusion- and flow-based policies. Through comprehensive benchmarks across diverse manipulation tasks, DA-SIP achieves **2.6–4.4× reduction in total computation time** while maintaining task success rates comparable to fixed maximum-computation baselines. By implementing adaptive computation within this framework, DA-SIP transforms generative robot controllers into efficient, task-aware systems that intelligently allocate inference resources where they provide the greatest benefit.

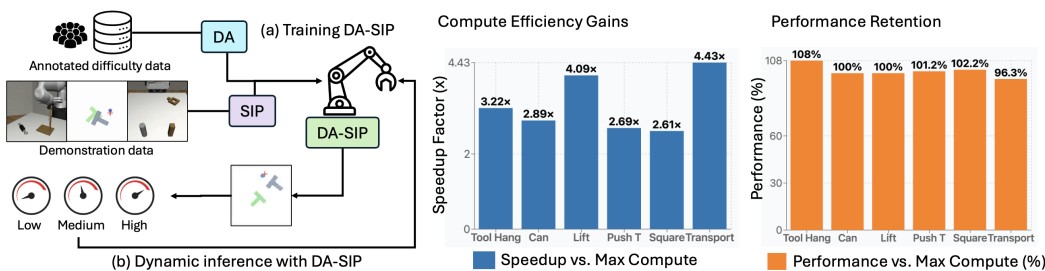

Figure 1: Overview of the DA-SIP framework with computational efficiency gains and performance retention

## 1 Introduction

Diffusion [1, 2] and flow-matching [3–5] models have recently shown strong performance in imitation learning for robotic policies, producing long-horizon action plans. By iteratively denoising trajectories (diffusion) or integrating probability-flow ODEs (flows), they capture the multi-modal distributions present in human demonstrations. These policies can be coupled with Vision-Language Models

39th Conference on Neural Information Processing Systems (NeurIPS 2025).

(VLMs) to form *Vision-Language-Action models (VLAs)* that map natural language goals to low-level control [6, 7].

Despite their success, existing controllers adopt a **uniform inference budget**: every control cycle executes the same solver, step schedule, and interpolant—all chosen to handle the **hardest** subtasks. This wastes computation on easy subtasks while ignoring a key advantage of flow-based generative models: the solver type, step count, and ODE/SDE formulation can all be adjusted at test time without retraining.

Large language models already employ adaptive computation: they use more reasoning steps for difficult queries and fewer for simple ones [8]. Physical manipulation tasks exhibit similar heterogeneity: coarse motions (e.g., moving an arm in free space) can be planned with minimal computation, while sub-millimeter placements demand more computational resources for precision.

To address this gap, we develop *Difficulty-Aware Stochastic Interpolant Policy (DA-SIP)*, which uses difficulty classification to dynamically allocate computational resources within the stochastic interpolant framework. Unobstructed object approaches receive minimal computational budgets, while precision manipulations command substantially higher budgets to ensure optimal performance.

TThis adaptive mechanism incorporates a dataset of demonstrations annotated with subtask difficulty levels. These annotations are used to train three complementary difficulty classifiers: (i) a lightweight CNN designed for efficiency, (ii) a few-shot vision-language model (VLM) prompted with exemplar images, and (iii) a few-shot VLM fine-tuned on our collected data. During each control cycle, the selected classifier evaluates the current RGB-D observation and predicts a difficulty level.

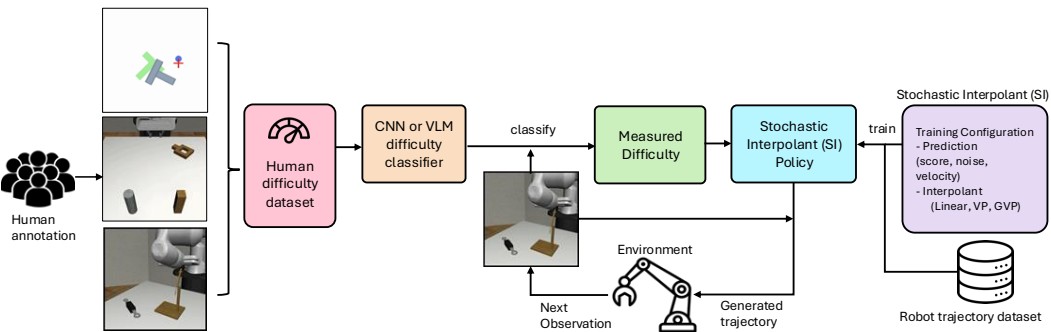

Figure 2: High-level overview of our difficulty-aware stochastic interpolant policy (DA-SIP) framework. During training we choose (1) a prediction target (noise, score, or velocity), (2) an interpolant (Linear, VP, GVP, *etc.*), and this configuration yields a single generative policy network that can perform both ODE and SDE integration. At inference time, a learned difficulty classifier adaptively selects an inference configuration triple $\langle$step count, solver type, ODE/SDE formulation$\rangle$ based on the current state—enabling context-dependent "System 1 vs. System 2" compute that maximizes success while minimizing latency.

To implement this adaptive computation effectively, we ground our approach in the *stochastic interpolant* (SI) framework [9], which unifies flow-based (ODE) and diffusion-based (SDE) generative modeling. This formulation enables a critical capability: translating difficulty levels into optimized inference configurations. Specifically, each difficulty level determines an inference configuration $\langle$step count, solver type, ODE/SDE integration$\rangle$, allowing the system to dynamically adjust computational resources (Fig. 2).

### Contributions.

1. **SI-grounded generative policy framework.** We unify diffusion and flow policies under stochastic interpolants, *exposing* a spectrum of training and inference choices in continuous time.

2. **Difficulty-aware adaptive compute.** The difficulty classifier selects solver type and step count using a lightweight CNN, few-shot VLM, or fine-tuned VLM, yielding a $2.6$–$4.4\times$ compute reduction without sacrificing success rate.

3. **Comprehensive benchmark study.** We demonstrate that our difficulty-guided approach reduces task completion time while matching the success rate of the maximum-compute baseline, and we analyze the resulting latency-accuracy trade-off.

## 2 Related Work

**Generative models**   Modern generative modeling techniques include diffusion models [1, 2, 10], flow matching [3–5], and stochastic interpolants that unify both frameworks [9, 11]. Diffusion models iteratively denoise data using score-matching objectives, while flow matching directly trains neural ODEs along flexible interpolation paths [3]. Recent distillation techniques further enhance computational efficiency [12–15]. The *stochastic interpolant* framework **uncouples the interpolation trajectory from the choice of deterministic (ODE) or stochastic (SDE) dynamics**, clarifying the design space and enabling independent control over sample quality, speed, and robustness.

**Generative modeling for decision making**   Diffusion Policy adapts diffusion-based generative modeling to learn robot control policies from human demonstrations [16]. Generating action sequences rather than single actions has demonstrated strong performance [17]. Flow-matching policies model the flow from noise to expert behavior through continuous transformations [18]. Moreover, distillation methods for control policies have been introduced [19, 20]. AdaFlow measures the multimodality of the action space in imitation learning by estimating the variance of possible actions at each state [21]. While it introduces adaptation by measuring multimodality in the action space, it addresses only a single dimension of state complexity-variance. In contrast, our approach recognizes multiple dimensions of state difficulty and provides a more comprehensive adaptation framework built on the stochastic interpolant framework, adjusting the full inference triplet: step count, solver type, and ODE/SDE integration mode.

**LLMs and VLMs for Robotic Control**   Recent work has integrated large language and vision-language models into robot control, creating Vision-Language-Action (VLA) models that map natural language and visual observations to control actions [7, 22, 23]. SayCan [24] grounds LLMs by constraining language outputs to feasible robot skills, while Inner Monologue [25] enables closed-loop reasoning where environmental feedback guides task execution. Our difficulty-aware approach in DA-SIP implements adaptive computation at the action generation level, conceptually similar to how LLMs allocate more "thinking" steps to complex queries [8], but applied to the visual control domain.

## 3 Methods

### 3.1 Generating Flow Policies with Stochastic Interpolants

Flow- and diffusion-based policies transform standard Gaussian noise $\varepsilon \sim \mathcal{N}(\mathbf{0}, \mathbf{I})$ into a sequence of robot actions $\mathbf{x}_* \sim p(\mathbf{x} \mid \mathbf{o})$, where $\mathbf{o}$ are observations. Here we do so within the stochastic interpolant framework [9]. To this end, we first introduce the **stochastic interpolant**

$$\mathbf{I}_t = \alpha_t \mathbf{x}_* + \sigma_t \varepsilon, \qquad t \in [0, 1], \tag{1}$$

where $\alpha_t$ decreases from $\alpha_0 = 1$ to $\alpha_1 = 0$ and $\sigma_t$ increases from $\sigma_0 = 0$ to $\sigma_1 = 1$, thereby interpolating between the action sequence $\mathbf{I}_0 = \mathbf{x}_*$ at $t = 0$ and pure noise $\mathbf{I}_1 = \varepsilon$ at $t = 1$.

Associated with the stochastic interpolant (1) we define the **velocity field**

$$\mathbf{v}(\mathbf{x}, t, \mathbf{o}) = \mathbb{E}\big[\dot{\mathbf{I}}_t \mid \mathbf{I}_t = \mathbf{x}, \mathbf{o}\big] \tag{2}$$

where $\mathbb{E}[\cdot \mid \mathbf{I}_t = \mathbf{x}, \mathbf{o}]$ denotes the expectation over $\mathbf{x}_*, \varepsilon$ conditional on $\mathbf{I}_t = \mathbf{x}$ and $\mathbf{o}$ fixed. This velocity can be efficiently learned by minimizing the loss

$$L_v[\hat{\mathbf{v}}] = \mathbb{E}\big[|\hat{\mathbf{v}}(\mathbf{I}_t, t, \mathbf{o}) - \dot{\mathbf{I}}_t|^2\big], \tag{3}$$

where the expectation is taken over $\mathbf{x}_*, \varepsilon, \mathbf{o}$ and $t \sim U(0, 1)$.

The **score** of the probability density of the interpolant (i.e. the gradient of the logarithm of this density) is given, via Stein's relation, as

$$\mathbf{s}(\mathbf{x}, t, \mathbf{o}) = -\sigma_t^{-1} \mathbb{E}\big[\varepsilon \mid \mathbf{I}_t = \mathbf{x}, \mathbf{o}\big] \tag{4}$$

which can be learned by minimizing the loss

$$L_s[\hat{\mathbf{s}}] = \mathbb{E}\big[|\hat{\mathbf{s}}(\mathbf{I}_t, t, \mathbf{o}) + \sigma_t^{-1}\boldsymbol{\varepsilon}|^2\big], \tag{5}$$

Alternatively, the score can be expressed in terms of the velocity (2) and *vice-versa* as

$$\mathbf{s}(\mathbf{x}, t, \mathbf{o}) = \sigma_t^{-1}\frac{\alpha_t \mathbf{v}(\mathbf{x}, t, \mathbf{o}) - \dot{\alpha}_t \mathbf{x}}{\dot{\alpha}_t \sigma_t - \alpha_t \dot{\sigma}_t} \quad \Leftrightarrow \quad \mathbf{v}(\mathbf{x}, t, \mathbf{o}) = \frac{\alpha_t}{\alpha_t}\mathbf{x} + \frac{\sigma_t(\dot{\alpha}_t \sigma_t - \alpha_t \dot{\sigma}_t)}{\alpha_t}\mathbf{s}(\mathbf{x}, t, \mathbf{o}) \tag{6}$$

These equations can be obtained by combining (2) and (4) with the relation $\mathbf{x} = \mathbb{E}\big[\mathbf{I}_t \mid \mathbf{I}_t = \mathbf{x}, \mathbf{o}\big]$, decomposing the conditional expectation of the sums into a sum of conditional expectations, and solving for $\mathbf{s}(\mathbf{x}, t, \mathbf{o})$ or $\mathbf{v}(\mathbf{x}, t, \mathbf{o})$.

Finally, we introduce the **reverse-time SDE**:

$$\mathrm{d}\mathbf{X}_t = \Big[\mathbf{v}(\mathbf{X}_t, t, \mathbf{o}) - \tfrac{1}{2}w_t s(\mathbf{X}_t, t, \mathbf{o})\Big]\mathrm{d}t + \sqrt{w_t}\,\mathrm{d}\bar{\mathbf{W}}_t, \qquad \mathbf{X}_1 = \boldsymbol{\varepsilon} \tag{7}$$

where $w_t \geq 0$ is a diffusion coefficient that can be adjusted post-training and $\bar{\mathbf{W}}_t$ is a reverse Wiener process. The probability distribution of the solution to (7) conditional on $\mathbf{o}$ coincides at all times with the probability distribution of the interpolant conditional on $\mathbf{o}$. In particular, the distribution of $\mathbf{X}_0$ is the target distribution $p(\mathbf{x} \mid \mathbf{o})$, indicating that (7) can be used as a generative model of sequence of robot actions given observations. The unconditional case is recovered by dropping the conditioning on $\mathbf{o}$.

In practice, we instantiate the equations above using **three stochastic interpolants** with different choices of $\alpha_t$ and $\sigma_t$:

| **Linear** | **Variance-Preserving (VP)** | **generalized VP (GVP)** |
|:---:|:---:|:---:|
| $\alpha_t = t,$ | | $\alpha_t = \sin\big(\frac{\pi}{2}t\big),$ |
| $\sigma_t = 1 - t.$ | $\alpha_t = \sqrt{1 - \exp\big(-\int_0^t \beta_s\,\mathrm{d}s\big)},$ | $\sigma_t = \cos\big(\frac{\pi}{2}t\big).$ |
| | $\sigma_t = \exp\big(-\frac{1}{2}\int_0^t \beta_s\,\mathrm{d}s\big).$ | |

These stochastic interpolants correspond to different ways to interpolate between noise and action sequences. Linear interpolation provides a direct, computationally efficient path; VP interpolation, up to time rescaling, recovers the formulation used in diffusion models; and GVP's trigonometric formulation ensures analytical normalization ($\alpha_t^2 + \sigma_t^2 = 1$).

## 3.2 Difficulty Classification Methods

We explore three approaches for classifying subtask difficulty from RGB-D observations, ranging from lightweight neural networks to vision-language models.

1. **Lightweight CNN Classifier.** A ResNet-18 backbone with a classification head processes RGB-D observations to predict difficulty levels. This supervised approach is trained on 300 annotated images per task (from a total dataset of ~2000 timesteps per task), providing efficient real-time classification with minimal computational overhead (~20ms per inference).

2. **Few-Shot VLM Classifier.** This zero-training approach leverages pre-trained vision-language models prompted with 1–3 exemplar images per difficulty category. The VLM classifies the current RGB-D observation by comparing it to the provided examples, requiring no task-specific training but incurring higher inference latency (~500–1000ms).

3. **Fine-Tuned VLM Classifier.** We fine-tune a pre-trained vision-language model on our annotated difficulty dataset (300 training images per task). This approach combines the strong representational capacity of VLMs with improved inference speed (~300–400ms) and achieves the highest classification accuracy across tasks.

## 3.3 Adaptive Computation Allocation

At each control step, the difficulty classifier outputs a difficulty level $d(\mathbf{o}_t) \in \{1, 2, 3\}$ for the current observation $\mathbf{o}_t$. This difficulty level is mapped to an *inference configuration triple*

$$(N_t, \text{ solver}_t, \text{ type}_t) = \mathcal{M}(d(\mathbf{o}_t)),$$

where $N_t \in \{5, 10, 20\}$ is the number of integration steps, $\text{solver}_t \in \{\text{Euler, Heun, RK4}\}$ specifies the numerical integrator, and $\text{type}_t \in \{\text{ODE, SDE}\}$ determines the integration mode. The mapping $\mathcal{M}$ is determined empirically by evaluating performance-efficiency trade-offs on a validation set. The configuration triple is then used by the Stochastic Interpolant Policy (SIP) to generate the next action sequence:

$$\mathbf{x} \sim \text{SIP}(\mathbf{o}_t; N_t, \text{solver}_t, \text{type}_t).$$

**Computational efficiency.** For our three-level difficulty system, we map difficulty levels to configurations that balance performance and compute cost:

- **Easy** ($d = 1$): ($N = 5, \text{Euler, ODE}$) — minimal computation for coarse motions
- **Medium** ($d = 2$): ($N = 10, \text{Heun, ODE}$) — moderate resources for approach phases
- **Hard** ($d = 3$): ($N = 20, \text{RK4, SDE}$) — maximum computation for precision tasks

This adaptive allocation enables the system to use minimal resources for simple subtasks while reserving expensive high-fidelity integration for challenging manipulation phases.

# 4 Experiments and Results

We evaluate our adaptive flow policy framework across various robotic manipulation tasks to demonstrate its versatility and effectiveness.

## 4.1 Simulation Environments

Our evaluation spans diverse simulation environments:

- **RoboMimic**: Benchmark suite for imitation learning in manipulation (Can, Lift, Square, Tool Hang tasks) [26]
- **Block Push**: A non-prehensile manipulation task (from the Fetch suite) in which a 7-DoF arm must push a cubic block across a tabletop to a target pose, requiring precise contact planning and continuous control [27]
- **Push-T**: Precision manipulation task requiring continuous control [28]
- **Kitchen**: Multi-stage environment with sequential task completion [29]
- **Multimodal Ant**: Locomotion tasks requiring complex coordination [30]

These environments represent a spectrum of complexity, from simple locomotion to intricate multi-object manipulation.

## 4.2 Optimizing Solver Configurations for Diverse Robotic Tasks

For uniformity and fair comparison across baselines, we focus on *velocity* prediction, although our framework also supports *score* prediction as an alternative training target [2, 4]. Our experiments reveal that different robotic tasks require distinct solver configurations to achieve optimal performance, as summarized in Table 1. Our adaptive computation mechanism draws inspiration from adaptive inference in neural networks [8] and builds upon recent advances in flow-based policy learning [18, 20, 31–33].

Our systematic evaluation revealed a key finding: there is no single best configuration that universally outperforms others across all environments. Initially, we hypothesized that evaluating all possible training and inference configurations of the stochastic interpolant framework would yield one optimal configuration outperforming all others across tasks. Instead, we discovered that different tasks require fundamentally different configurations to achieve optimal performance.

**Simple manipulation tasks.** Simple tasks like Can and Lift achieve near-optimal performance with minimal computation, with even 1-step configurations reaching 99–100% success. Even single-step flow model inference without distillation achieves the highest maximum performance, indicating that additional inference computation is wasted.

Table 1: Task characterisation and optimal configurations.

| Task | PushT | Block Push | Tool Hang | Multimodal Ant | Lift | Can | Transport | Square |
|---|---|---|---|---|---|---|---|---|
| Solver | Heun | Heun | Euler | Euler | Euler | Euler | Euler | Euler |
| Interpolation | VP | GVP | VP | VP | Linear | Linear | VP | VP |
| Integration | SDE | SDE | ODE | ODE | ODE | ODE | SDE | SDE |
| Inference steps | 100 | 100 | 50 | 25 | 1 | 1 | 100 | 100 |
| Last step | Tweedie | Tweedie | None | None | None | None | Euler | Euler |
| Success rate (%) | **92.6** | **22.8** | **38.1** | **45.3** | **100.0** | **100.0** | **85.0** | **95.2** |

Each configuration is trained for 5,000 epochs with three random training seeds. The model uses a U-Net transformer architecture. During evaluation, we use three inference seeds per training seed (nine runs per configuration). Checkpoints are saved every 50 epochs, and we report the mean success rate of the final 10 checkpoints, with each checkpoint evaluated over 50 independent episodes. Training is performed on NVIDIA L40S GPUs.

**Precision manipulation tasks.** Push-T and Block Push demonstrate strong dependency on the integration method. These tasks require substantially more computational resources because millimeter-precision actions produce meaningful differences in success rates. Heun approximation outperforms Euler by a slight margin ($\pm 0.1\%$) consistently across different training and inference seeds, highlighting the importance of accurate numerical integration for high-precision control tasks.

**Exploratory manipulation tasks.** Most notably, Tool Hang and Multimodal Ant tasks achieve peak performance at intermediate step counts. Increasing from the optimal step count (50 for Tool Hang, 25 for Multimodal Ant) to 100 steps actually decreases performance by 0.5–2.9%, suggesting that excessive computation can be counterproductive. We attribute this phenomenon to the need for controlled stochasticity in exploratory manipulation—similar to threading a needle, where some variation aids success but excessive randomness becomes detrimental.

**Transport and placement tasks.** Square and Transport tasks represent a middle ground, requiring neither the minimal computation of simple tasks nor the extreme precision of complex ones. While they involve multiple sequential actions, they don't demand the millimeter-level precision or carefully controlled stochasticity of the more challenging environments.

This diversity in optimal configurations reinforces our understanding that different robotic tasks have distinct computational requirements, motivating adaptive methods that adjust computational resources based on task characteristics.

## 4.3 Categorizing Tasks by Difficulty

In all tasks, we identify distinct phases through which the robot progresses. When the robot is initialized far from interaction objects, we categorize this as the **Initial** state (I). As the robot approaches within approximately 10 cm of an object, we designate this as the **Near** state (N). Subsequently, the robot engages in one of three interaction types: **Grasping** (G), where the robot grasps and places an object; **Stochastic attempts** (S), which require exploratory variability for precise placement; or **Continuous pushing** (C), where the robot pushes objects without grasping.

By categorizing tasks based on manipulation type to assign an appropriate difficulty level, we avoid the need to find optimal configurations for each individual task, which would be impractical. We use a standardized categorization system that can be applied uniformly across all tasks. This one-time categorization eliminates any additional per-task tuning at deployment.

## 4.4 Difficulty Classification Performance

We developed a methodology to reliably categorize states across our environments. We collected annotations from 8 different human labelers, accumulating approximately 20,000 labeled states across the six difficulty categories (I, N, G, S, C, E). Our ablation studies show that 300 images per task prove sufficient (Appendix E.1). For each state, we assigned the final label based on majority voting, selecting the category with the highest frequency of votes among annotators. This consensus-driven

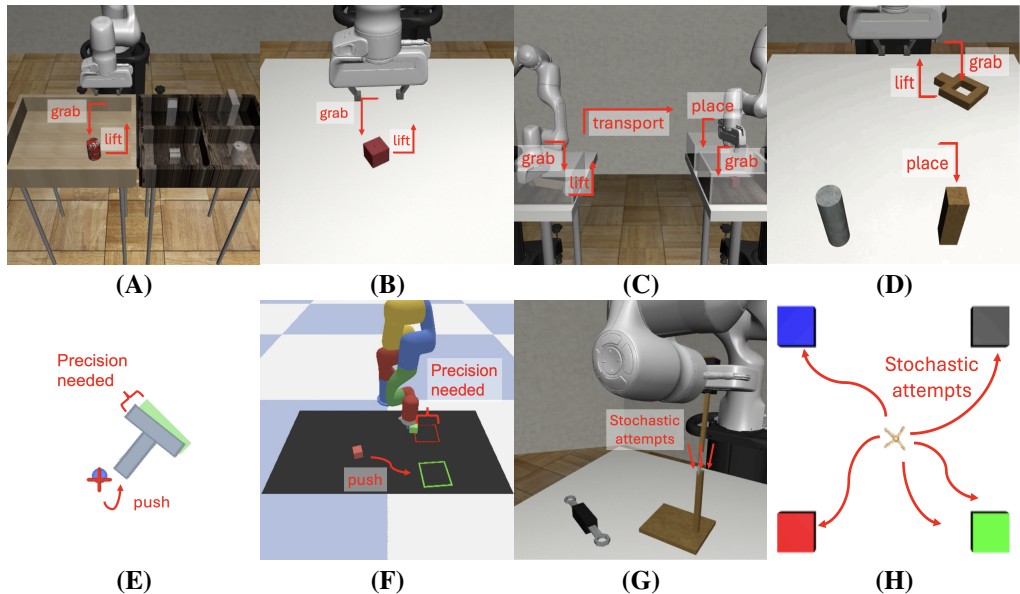

Figure 3: **Robot manipulation tasks across complexity categories. (A-B)** Simple manipulation tasks (*Can* and *Lift*) require minimal computational steps while maintaining high success rates. **(C-D)** Transport and placement tasks (*Transport* and *Square*) show greater sensitivity to configuration choices, representing medium-complexity challenges. **(E-F)** Precision manipulation tasks (*Push T* and *Block Push*) demonstrate significant benefits from Heun integration and variance-preserving diffusion models for fine-grained control. **(G-H)** Exploratory manipulation tasks (*Tool Hang* and *Multimodal Ant*) highlight that adaptive resource allocation outperforms maximum computation, addressing complex challenges with varying difficulty levels.

approach helped mitigate individual biases while establishing a comprehensive ground truth dataset. Additionally, we selected a small representative subset (1–3 images per category) for few-shot prompting with vision-language models (VLMs). We fine-tuned several VLMs on our dataset and selected the best-performing model for classification (Table 2).

(a) Lightweight CNN accuracy

| Env. | Acc. (%) |
|---|---|
| Can | 88.9 |
| Lift | 92.8 |
| Push-T | 94.6 |
| Square | 87.4 |
| Tool Hang | 81.3 |
| Transport | 91.5 |

(b) Few-shot VLM average

| VLM | Avg. (%) |
|---|---|
| LLAVA | 25.0 |
| Gemma-3 | 23.0 |
| Phi-4 | 24.6 |
| **Qwen2.5** | **45.3** |
| **Qwen2.5 (Fine-tuned)** | **53.9** |

(c) Qwen-VL 2.5 per-env.

| Env. | Few-shot (%) | Fine-tuned (%) |
|---|---|---|
| Can | 37.5 | 47.5 |
| Lift | 83.3 | 86.7 |
| Push-T | 42.0 | 56.7 |
| Square | 26.0 | 52.0 |
| Tool Hang | 50.0 | 40.0 |
| Transport | 30.0 | 40.0 |

Table 2: Difficulty classifier accuracy. We test different models [34–36]. For CNN, our ablation studies in Appendix E.1 show that approximately 300 images per task achieve near-plateau performance, requiring only about 15 minutes of annotation effort per task. The CNN classifier also demonstrates robustness to sensor noise (Appendix E.4). For the few-shot VLM, the impact of varying numbers of exemplar images is described in Appendix E.3.

Our lightweight CNN classifier demonstrated the strongest performance across all environments in both accuracy and inference time. Among the tested VLMs, Qwen-VL 2.5 achieved the highest average accuracy (45.3%) for few-shot multi-image prompting. We fine-tuned this VLM for 8 epochs, and fine-tuning with a minimal number of images further boosted accuracy (Table 2).

## 4.5 Assigning inference triplets based on difficulty label

We assign minimal computational resources (1 inference step, Euler solver, ODE integration) to Initial and End states, as these phases require basic positioning in unobstructed space. The Near category receives 5 inference steps with Euler solver and ODE integration, sufficient for approaching objects while avoiding collisions. For Grabbing interactions, we allocate 10 inference steps with Euler and ODE, as these do not require maximal computational resources. Stochastic attempts benefit from controlled variability, so we assign 50 inference steps with SDE integration, which produced optimal performance in the Tool Hang task. Finally, Continuous pushing, requiring the highest precision, receives 100 inference steps with Heun solver and SDE integration to achieve millimeter-precise control.

| Category | Characteristics | Inference Steps | Solver | Integration |
|---|---|---|---|---|
| Initial state (I) | Robot positioned away from targets | **1** | **Euler** | **ODE** |
| Near (N) | Approaching within 30cm of targets | **5** | **Euler** | **ODE** |
| Grabbing (G) | Grasping and placing objects | **10** | **Euler** | **ODE** |
| Stochastic attempts (S) | Precise alignment requiring variability | **50** | **Euler** | **SDE** |
| Continuous pushing (C) | Sustained precise manipulation | **100** | **Heun** | **SDE** |
| End state (E) | Objectives achieved | **1** | **Euler** | **ODE** |

Table 3: State difficulty categories and their computational requirements

## 4.6 Adaptive Computation Efficiency

Based on our empirical results across task categories, we mapped state difficulty to optimal computational configurations for each task phase.

Overall, our DA-SIP achieved substantial reductions in task completion time compared to maximum computation baselines: an average 3.3-fold reduction with adaptive CNN, 2.7-fold with few-shot VLM, and 2.3-fold with fine-tuned VLM. Performance remained robust across most methods, with average performance differences of only 1.3%, 4.7%, and 1.8% from maximum computation, respectively. Notably, the fine-tuned VLM achieved an excellent balance between the computational efficiency of CNN-based classification and the deployment flexibility of few-shot VLM approaches. With minimal training data, it achieved consistent performance across environments, even demonstrating significant performance improvements (up to 8% in Tool Hang) while maintaining the adaptability benefits of vision-language models. These results confirm that dynamically allocating computational resources based on state difficulty substantially improves efficiency while preserving task performance, with fine-tuned VLMs offering a particularly appealing balance between reliability and adaptability.

| Simulation | SIP | | DA-SIP | | |
| | Min Compute | Max Compute | Lightweight CNN | Few-shot VLM | Fine-tuned VLM |
|---|---|---|---|---|---|
| Tool Hang | 91.02 | 342.13 | 106.37 (**-3.22x**) | 150.32 (**-2.28x**) | 139.53 (**-2.45x**) |
| Square | 25.86 | 146.68 | 56.20 (**-2.61x**) | 89.04 (**-1.65x**) | 108.76 (**-1.35x**) |
| Transport | 73.04 | 342.54 | 77.36 (**-4.43x**) | 178.29 (**-1.92x**) | 163.38 (**-2.10x**) |
| Push T | 1.44 | 121.89 | 45.35 (**-2.69x**) | 21.96 (**-5.55x**) | 45.34 (**-2.69x**) |
| Lift | 57.04 | 263.28 | 64.31 (**-4.09x**) | 101.00 (**-2.61x**) | 89.50 (**-2.94x**) |
| Can | 63.70 | 214.81 | 74.32 (**-2.89x**) | 94.94 (**-2.26x**) | 86.44 (**-2.49x**) |

Table 4: Time comparison across different simulations showing computational efficiency gains. Values indicate computation time (seconds), with fold reduction relative to maximum compute shown in parentheses as negative multiples (e.g., -3.2x indicates 3.2× faster than maximum compute). The adaptive configurations achieve notable efficiency, with minimum compute delivering up to **84.6×** reduction in the Push T environment, and Few-shot VLM offering up to **5.6×** speedup. A detailed timing breakdown is provided in Appendix E.2.

| Simulation | SIP | | DA-SIP | | |
|---|---|---|---|---|---|
| | Min Compute | Max Compute | Lightweight CNN | Few-shot VLM | Fine-tuned VLM |
| Tool Hang | 25% | 25% | 27% (**+2%**) | 33% (**+8%**) | 33% (**+8%**) |
| Square | 88% | 90% | 92% (**+2%**) | 88% (**-2%**) | 89% (**-1%**) |
| Transport | 62% | 80% | 77% (**-3%**) | 62% (**-18%**) | 79% (**-1%**) |
| Push T | 81% | 86% | 87% (**+1%**) | 86% (**0%**) | 87% (**+1%**) |
| Lift | 100% | 100% | 100% (**0%**) | 100% (**0%**) | 100% (**0%**) |
| Can | 98% | 99% | 99% (**0%**) | 99% (**0%**) | 99% (**0%**) |

Table 5: Performance comparison showing generally minimal success rate degradation despite computational savings. Values indicate success rate, with percentage difference relative to maximum compute shown in parentheses. Most configurations maintain performance close to maximum compute, with three environments (Push T, Lift, Can) showing no performance loss with the few-shot VLM approach. The CNN classifier maintains strong performance in Transport (-3% from maximum), while few-shot VLM shows a more substantial performance trade-off (-18%) in this environment.

## 5  Conclusion

We introduced **DA-SIP**, a *Difficulty-Aware Stochastic Interpolant Policy* that brings adaptive test-time computation to generative robot control. By casting diffusion and flow policies in the stochastic interpolant framework and attaching a difficulty classifier, DA-SIP selects the solver depth, interpolant, and ODE/SDE mode on the fly. Across our simulated manipulation suite, this approach cuts total computation by $2.6$–$4.4\times$ while matching the task success rates of fixed, uniform-budget baselines.

This work contributes (i) a unified SI formulation that exposes a rich design space for generative policies, (ii) an adaptive inference system that dynamically selects computational parameters without modifying the underlying policy, and (iii) empirical evidence demonstrating significant computational savings without performance degradation. While our primary evaluation focuses on simulation environments, preliminary experiments with real robot data show that SIP achieves lower mean squared error than diffusion policy on action prediction (Appendix E.5), suggesting promise for physical robot deployment. Future directions include applying this adaptive inference framework to larger robotics foundation models and coupling it with safety monitors for contact-rich tasks. Together, these advances move generative controllers toward efficient, deployable autonomy in resource-constrained physical environments.

## Acknowledgments

We thank the authors of Stochastic Interpolant and the SiT for their paper and codebase, which served as the foundation of our research [9, 11]. We are also grateful for the authors of the Diffusion Policy codebase [16], which was instrumental for our implementation.

We thank Sanghyun Woo for valuable early discussions on research methodology and comparative analysis, and Nur Muhammad Mahi Shafiullah for insightful discussions on robotics experiment design and ensuring fair comparisons with baseline methods. We thank Lerrel Pinto and Mark Goldstein for helpful feedback on the draft and experiments. We are grateful to the annotators, including authors, Manish Chowdhury, Elvis Fiador, Jiwon Hwang, Yeonwoo Kim, Bruna Sampaio, and Aditya Singh for their assistance with data annotation. We thank Nanye Ma for helpful discussions about the SiT concepts and codebase, and for valuable feedback on our research approach.

This work was supported by compute resources provided by NYU IT High Performance Computing resources and services.

We also thank the anonymous reviewers for their constructive feedback that helped improve this paper.

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

# Appendix A   Implementation Details

This appendix provides key implementation details of our DA-SIP framework.

## A.1   Stochastic Interpolant Policy Implementation

Table 6: Stochastic interpolant policy hyperparameters

| Parameter | Value | Parameter | Value |
|---|---|---|---|
| Optimizer | AdamW | Weight decay | 1e-6 |
| Learning rate | 1e-4 | Schedule | Cosine decay |
| Batch size | 256 | Gradient clipping | 1.0 |
| Training epochs | 5,000 | Checkpointing | Every 50 epochs |
| Loss function | MSE | EMA rate | 0.9999 |
| Prediction targets | Velocity/Score/Noise | LR Scheduler | Cosine decay |
| Interpolants | Linear/VP/GVP | warmup steps | 500 |

## A.2   Lightweight CNN Classifier

Our CNN classifier uses three convolutional blocks followed by fully connected layers:

Table 7: Lightweight CNN architecture and training parameters

| Architecture | | Training Parameters | |
|---|---|---|---|
| Input | RGB image $32 \times 32 \times 3$ | Optimizer | Adam |
| Conv1 | 32 filters, $3 \times 3$, BatchNorm, ReLU | Learning rate | 1e-3 |
| MaxPool1 | $2 \times 2$ stride 2 | Batch size | 64 |
| Conv2 | 64 filters, $3 \times 3$, BatchNorm, ReLU | Epochs | 100 |
| MaxPool2 | $2 \times 2$ stride 2 | Early stopping | patience=15 |
| Conv3 | 128 filters, $3 \times 3$, BatchNorm, ReLU | Loss function | Categorical cross-entropy |
| MaxPool3 | $2 \times 2$ stride 2 | Class weighting | Inverse frequency |
| Flatten | 2048 units | Validation split | 20% |
| FC1 | 256 units, ReLU, Dropout(0.5) | Augmentation | Flip, rotation, color jitter |
| FC2 | 6 units, Softmax | | |

We apply histogram equalization per color channel and use weighted random sampling to handle class imbalance with:

$$\text{weights} = \frac{n\_\text{samples}}{n\_\text{classes} \times \text{class\_counts}} \tag{8}$$

## A.3   Few-shot VLM Classification

We use few-shot in-context learning with the following configuration:

- **Models evaluated:** Qwen-VL 2.5, LLAVA-Next, Gemma 3
- **Method:** Present reference images with labels alongside target image
- **Prompt:** "simply pick the label from the sample images you see in prompt and tell me which one is closest to what you see"
- **Category mapping:**

| i | initialize | s | stochastic exploration |
|---|---|---|---|
| n | near | e | end state |
| g | grab | c | continuous pushing |

## A.4 Fine-tuned VLM Implementation

Table 8: VLM fine-tuning configuration

| Model Configuration | Value | Training Parameters | Value |
|---|---|---|---|
| Base model | Qwen2.5-VL-7B-Instruct | Learning rate | 2e-5 |
| Quantization | 8-bit, NF4 type | Batch size | 32 |
| Language LoRA rank | 16 | Epochs | 12 |
| Vision LoRA rank | 8 | Optimizer | AdamW |
| LoRA alpha | 32 | Weight decay | 0.01 |
| Vision LoRA alpha | 16 | Warmup ratio | 3% |
| LoRA dropout | 0.05 | Precision | FP16 |
| Target modules | q/k/v/o/gate/up/down proj | Gradient checkpointing | Enabled |
| Image resolution | 224×224 | Max sequence length | 200 tokens |

# Appendix B   Performance comparison

Table 9: Performance on the "Lift" task comparing DDPM, DDIM, and Linear (SI Policy with linear interpolation) under varying inference steps. We used Euler ODE approximator for trained models for SIP. **Note that this is without distillation for SIP**

| Inference Steps | Diffusion Policy (DDPM) | Diffusion Policy (DDIM) | SI Policy (ours) |
|---|---|---|---|
| 100 | **1.00** | **1.00** | **1.00** |
| 50 | **1.00** | 0.99 | **1.00** |
| 10 | 0.04 | 0.03 | **1.00** |
| 5 | 0.03 | 0.02 | **1.00** |
| 1 | 0.00 | 0.03 | **1.00** |

Table 10: Success rates on the "Can" manipulation task: Comparing conventional diffusion methods (DDPM, DDIM) with our SI Policy using linear interpolation across different inference step counts. All models employ Euler ODE approximation without distillation.

| Inference Steps | Diffusion Policy (DDPM) | Diffusion Policy (DDIM) | SI Policy (ours) |
|---|---|---|---|
| 100 | **1.00** | **1.00** | **1.00** |
| 50 | 0.97 | 0.98 | **1.00** |
| 10 | 0.00 | 0.00 | **1.00** |
| 1 | 0.00 | 0.00 | **1.00** |

Table 11: Representative **PushT** results with **VP** interpolation, using either **Euler** or **Heun**, in **ODE** or **SDE** mode, and varying numbers of steps. Performance (success rate) improves with more sophisticated solvers, showing that each configuration must be carefully tuned for optimal results.

| Interpolation | # Steps | ODE / SDE | Approximator | Final Step | Success Rate |
|---|---|---|---|---|---|
| VP | 10 | ODE | Euler | Euler | 0.786 |
| VP | 25 | ODE | Euler | Euler | 0.897 |
| VP | 50 | ODE | Euler | Euler | 0.912 |
| VP | 100 | ODE | Euler | Euler | 0.918 |
| VP | 100 | SDE | Euler | Linear | 0.918 |
| VP | 10 | ODE | Heun | Tweedie | 0.881 |
| VP | 25 | ODE | Heun | Tweedie | 0.916 |
| VP | 50 | ODE | Heun | Tweedie | 0.920 |
| VP | 100 | ODE | Heun | Tweedie | 0.922 |
| VP | 100 | SDE | Heun | Tweedie | **0.926** |

| Comparison of Diffusion Policy | | | |
|---|---|---|---|
| **Inference Steps** | **DP (DDPM)** | **DP (DDIM)** | **SI Policy (ours)** |
| 10 | 0.125 | 0.810 | 0.881 |
| 25 | 0.214 | 0.813 | 0.916 |
| 50 | 0.801 | 0.812 | 0.920 |
| 100 | 0.807 | 0.810 | **0.926** |

Table 12: **BlockPush** task success rate with GVP interpolation, comparing different approximators (Euler vs. Heun), ODE vs. SDE approaches, and final step methods. Results show that Heun with Tweedie correction achieves the highest success rates, while ODE performance scales with step count. The bottom section provides a direct comparison with standard Diffusion Policy.

| Interpolation | # Steps | ODE / SDE | Approximator | Final Step | Success Rate |
|---|---|---|---|---|---|
| GVP | 1 | ODE | Euler | - | 0.026 |
| GVP | 10 | ODE | Euler | - | 0.155 |
| GVP | 25 | ODE | Euler | - | 0.199 |
| GVP | 50 | ODE | Euler | - | 0.195 |
| GVP | 100 | ODE | Euler | - | 0.200 |
| GVP | 100 | SDE | Euler | Euler | 0.204 |
| GVP | 100 | SDE | Euler | Tweedie | 0.205 |
| GVP | 100 | SDE | Heun | Euler | 0.215 |
| GVP | 100 | SDE | Heun | Tweedie | **0.227** |
| **Comparison with Diffusion Policy** | | | | | |
| Method | | Configuration | | | Success Rate |
| Diffusion Policy (DDPM) | | 100 steps | | | 0.210 |
| SI Policy (ours) | | GVP, Heun, Tweedie, 100 steps | | | **0.227** |

Table 13: Performance on the **Transport** task with VP interpolation, comparing different numbers of steps, ODE vs. SDE approaches, and approximators. Results show that performance is highly dependent on sufficient step count, with SDE approaches achieving slightly better performance.

| Interpolation | # Steps | ODE / SDE | Approximator | Final Step | Success Rate |
|---|---|---|---|---|---|
| VP | 10 | ODE | Euler | - | 0.006 |
| VP | 25 | ODE | Euler | - | 0.758 |
| VP | 50 | ODE | Euler | - | 0.796 |
| VP | 100 | ODE | Euler | - | 0.784 |
| VP | 100 | SDE | Euler | Euler | **0.850** |
| VP | 100 | SDE | Heun | Tweedie | **0.850** |
| **Comparison with Diffusion Policy** | | | | | |
| **Diffusion Policy (DDPM)** | | | **SI Policy (ours)** | | |
| **0.852** | | | **0.850** | | |

Table 14: Performance on the **Square** task with VP interpolation, comparing different numbers of steps, ODE vs. SDE approaches, and approximators.

| Interpolation | # Steps | ODE / SDE | Approximator | Final Step | Success Rate |
|---|---|---|---|---|---|
| VP | 10 | ODE | Euler | – | 0.892 |
| VP | 25 | ODE | Euler | – | 0.950 |
| VP | 50 | ODE | Euler | – | 0.942 |
| VP | 100 | ODE | Euler | – | 0.926 |
| VP | 100 | SDE | Euler | Euler | **0.952** |
| **Comparison with Diffusion Policy** | | | | | |
| **Inference Steps** | | **DP (DDPM)** | | **DP (DDIM)** | **SI Policy (ours)** |
| 10 | | 0.000 | | 0.928 | 0.892 |
| 50 | | 0.916 | | 0.943 | 0.942 |
| 100 | | 0.944 | | **0.962** | 0.952 |

Table 15: Tool Hang performance with VP interpolation across different configurations (averaged across 3 seeds)

| Interpolation | # Steps | ODE/SDE | Solver | Success Rate |
|:---:|:---:|:---:|:---|:---:|
| VP | 1 | ODE | Euler | 0.000 |
| VP | 10 | ODE | Euler | 0.115 |
| VP | 25 | ODE | Euler | 0.345 |
| **VP** | **50** | **ODE** | **Euler** | **0.381** |
| VP | 100 | ODE | Euler | 0.351 |
| VP | 10 | ODE | Heun | 0.342 |
| VP | 25 | ODE | Heun | 0.330 |
| VP | 50 | ODE | Heun | 0.371 |
| VP | 100 | ODE | Heun | 0.364 |
| VP | 100 | SDE | Euler | 0.357 |
| VP | 100 | SDE | Heun | 0.368 |

| Comparison with Diffusion Policy | | | |
|:---:|:---:|:---:|:---:|
| **Step Count** | **DP (DDPM)** | **DP (DDIM)** | **SI Policy (ours)** |
| 1 | 0.000 | **0.482** | 0.000 |
| 10 | 0.000 | **0.484** | 0.342 |
| 25 | 0.000 | **0.486** | 0.371 |
| 50 | **0.534** | 0.490 | 0.381 |
| 100 | **0.484** | 0.480 | 0.368 |

*Note: Surprisingly, Diffusion Policy methods perform much better on Tool Hang than SI Policy consistently. DDPM achieves the highest overall success rate (0.534) at 50 steps, while DDIM shows remarkable consistency across all step counts, maintaining 0.48 success even at very low step counts.*

# Appendix C    Data Collection and Annotation

This appendix details our methodology for collecting and annotating the dataset of 20,000 labeled robot states used for training and evaluating our difficulty classification models.

We developed a systematic approach to collect data of robot states across various manipulation tasks. Our data collection protocol was as follows:

## C.1    Episode Recording and Frame Extraction

We recorded complete episodes across six simulation environments (Can, Lift, Push T, Square, Tool Hang, and Transport). We extracted frames at regular intervals (every 5th frame) with additional adaptive sampling to ensure representation of critical states (e.g., precise grasping moments). This approach yielded approximately 20,000 frames for annotation.

## C.2    Annotation Process

Eight volunteers participated in the annotation process. Each annotator received tutorial videos for how to annotate each image for each task. A custom web-based annotation platform was used to help them annotate and each volunteer is given a distinct username.

## C.3    Difficulty Categories

The following categories were used to classify robot states based on computational requirements:

1. **Initial (I)**: The robot is positioned away from targets and objects, performing gross positioning movements in free space. No precise control is needed.

2. **Near (N)**: The robot is approaching within approximately 10cm of a target object but has not yet initiated contact or grasping. Some care in motion planning is required.

3. **Grabbing (G)**: The robot is in the process of grasping an object, or has grasped an object and is moving it to a new location. Moderate precision is required.

4. **Stochastic (S)**: The robot is attempting a precise placement or alignment task that requires controlled variability (e.g., inserting a tool, threading a needle). High precision with some exploration is needed.

5. **Continuous (C)**: The robot is pushing or manipulating an object without grasping, requiring continuous fine control with millimeter precision (e.g., pushing a block along a specific path).

6. **End (E)**: Task objectives have been achieved, and the robot is in a terminal state or moving away from completed objectives.

## C.4    Dataset Finalization

To ensure reliability, each state was labeled by multiple annotators. For the final dataset, we assigned category labels using majority voting.

The most common boundary cases occurred between Near (N) and Grabbing (G) categories

These patterns reflect the continuous nature of difficulty transitions during task execution, with some states occupying boundary regions between categories.

The final dataset contains approximately 20,000 labeled states spanning all six difficulty categories across the different simulation environments, providing a comprehensive foundation for training our difficulty classifiers.

# Appendix D   Robustness/Efficiency Analysis

## D.1   Data Efficiency

To address the scalability concern of training the difficulty classifier, especially with the CNN approach, we conducted ablation studies to determine the minimum training data required to achieve comparable performance to our full dataset. We measured the accuracy of models trained on varying numbers of images, evaluating their predictions against majority-vote labels from human annotators.

Table 16: CNN classifier accuracy with varying training data sizes. Test set sizes shown in parentheses.

| Environment | Training Images | | | | | Test Set Size |
|---|---|---|---|---|---|---|
| | 100 | 200 | 300 | 500 | 2000 | |
| Square | 8% | 48% | 84% | 85% | 84% | 910 |
| Tool Hang | 2% | 23% | 19% | 29% | 39% | 1584 |
| Lift | 87% | 89% | 90% | 92% | 92% | 900 |
| Can | 2% | 2% | 85% | 83% | 88% | 520 |
| Transport | 1% | 1% | 36% | 36% | 37% | 1584 |
| **Average** | 20.0% | 32.6% | 62.8% | 65.0% | 68.0% | – |

The results show that 300 training images achieve near-plateau performance for Square, Can, and Lift, while Tool Hang peaks at around 200 images and Transport at 300 images. Therefore, approximately 300 images per task can be used as a reasonable baseline amount of training data needed to train an accurate classifier model, especially when training from scratch with architectures such as CNNs.

## D.2   Computational Overhead Analysis

We re-rolled out the policy for Tool Hang (88 iterations, per-iteration average):

Table 17: Per-cycle computational breakdown for different difficulty classifiers (in seconds)

| Component | CNN | Fine-tuned VLM |
|---|---|---|
| Classifier inference | 0.023 | 0.362 |
| SIP policy inference | 0.301 | 0.127 |
| Simulation execution | 0.977 | 0.991 |
| Data processing | 0.057 | 0.055 |
| **Total per cycle** | 1.357 | 1.535 |

## D.3   Few-shot VLM Learning

To investigate the impact of the number of exemplar images on VLM classification performance, we tested few-shot learning with varying numbers of images per category. The following results demonstrate the trade-off between classification accuracy and inference time.

Table 18: Few-shot VLM performance and inference time with varying exemplar images

| Environment | 1 img/cat | 2 img/cat | 3 img/cat |
|---|---|---|---|
| Square | 20% (0.55s) | 20% (0.89s) | 24% (1.26s) |
| Tool Hang | 16% (0.56s) | 32% (0.91s) | 52% (1.26s) |
| Lift | 57% (0.48s) | 83% (0.64s) | 83% (0.82s) |
| Can | 25% (0.52s) | 43% (0.75s) | 38% (1.05s) |
| Transport | 25% (0.50s) | 25% (0.72s) | 33% (1.05s) |
| **Average** | 28.6% (0.52s) | 40.6% (0.78s) | 46.0% (1.09s) |

## D.4 CNN Robustness to Noise

We also conducted additional environment evaluating the robustness of our CNN difficulty classifier under noisy sensor conditions. We added varying levels of gaussian noise to test images before classifying. This addresses the concerns about real-world deployment where sensor noise is unavoidable.

**Experimental setup:**

- Training samples per environment: 300
- Test set: 20% of data (stratified split)
- Random seed: 42 (ensures reproducible noise across runs)
- Noise levels: $\sigma \in \{0.0, 0.05, 0.15, 0.30, 2.0\}$

Table 19: CNN classifier accuracy (%) under different noise levels

| Environment | Clean ($\sigma$=0.0) | Low Noise ($\sigma$=0.05) | Medium Noise ($\sigma$=0.15) | High Noise ($\sigma$=0.30) | Extreme Noise ($\sigma$=2.0) |
|---|---|---|---|---|---|
| Square | 84.51 | 85.60 (+1.10) | 83.08 (-1.43) | 78.68 (-5.82) | 12.86 (-71.65) |
| Tool Hang | 20.52 | 20.27 (-0.25) | 19.95 (-0.57) | 19.63 (-0.88) | 1.83 (-18.69) |
| Lift | 91.67 | 90.67 (-1.00) | 89.00 (-2.67) | 81.89 (-9.78) | 75.11 (-16.56) |
| Can | 81.92 | 82.50 (+0.58) | 82.12 (+0.19) | 82.69 (+0.77) | 77.69 (-4.23) |
| Transport | 34.66 | 34.34 (-0.32) | 34.60 (-0.06) | 35.10 (+0.44) | 22.85 (-11.81) |
| **Average** | 62.66 | 62.68 (+0.02) | 61.75 (-0.91) | 59.60 (-3.06) | 38.07 (-24.59) |

This experiment demonstrates that our CNN models still maintain reasonable performance under high noise conditions ($\sigma$=0.30) with only 3.06% average accuracy drop. $\sigma$=2.0 represents extreme noise conditions beyond typical real-world scenarios.

## D.5 Real Robot Training Validation

To address concerns about sim-to-real transfer, we conducted an experiment with the 'real robot pushT' that has publicly available training data. We were able to train a diffusion policy and SIP with linear interpolation and velocity prediction and found a decreasing pattern of training action MSE, revealing that SIP can mimic the expert human demonstrator, which suggests it will be useful when deployed in the real world. Due to time constraints, we were unable to roll out actual robot trajectories.

**Experimental setup:**

- Dataset: Real robot PushT human demonstrations
- Policy configuration: SIP with linear interpolation and velocity prediction
- Baseline: Diffusion Policy with identical training setup
- Metrics: Training loss and action MSE over training steps

Table 20: Real robot training results

| Training Loss | | | Action MSE | | |
|---|---|---|---|---|---|
| Step | Diffusion Policy | SIP | Step | Diffusion Policy | SIP |
| 420 | 0.0760 | 0.183 | 420 | 0.0250 | 0.0136 |
| 2520 | 0.0238 | 0.085 | 2520 | 0.00582 | 0.00472 |
| 4614 | 0.0382 | 0.131 | 4614 | 0.00467 | 0.00294 |

