# OpenReview forum: "Dynamic Test-Time Compute Scaling in Control Policy: Difficulty-Aware Stochastic Interpolant Policy"
_NeurIPS.cc/2025/Conference — NeurIPS 2025 poster_

### Official Review · Reviewer_9NDW · 2025-06-08

**Clarity:** 3
**Significance:** 2
**Originality:** 2
**Rating:** 4
**Confidence:** 3

**Summary:**

**Summary**
This paper proposes **DA-SIP (Difficulty-Aware Stochastic Interpolant Policy)**, a novel framework for adaptive test-time computation in diffusion- and flow-based robotic controllers. The authors identify a key inefficiency in existing approaches: they use a fixed inference budget for all control steps, regardless of the underlying task complexity. To address this, DA-SIP introduces a **difficulty classifier** that dynamically assesses the complexity of a subtask from RGB-D inputs and selects the appropriate solver configuration, integration horizon, and ODE/SDE mode on the fly.

The method is grounded in the **stochastic interpolant (SI)** framework, which unifies diffusion and flow-based models and allows flexible inference configuration without retraining.
Through comprehensive benchmarking on robotic manipulation tasks, DA-SIP demonstrates **2.6–4.4× reductions in computation time** while maintaining **task success rates comparable to fixed-budget baselines**. The approach provides an efficient and scalable path forward for resource-aware generative robot policies.

**Questions:**

**Questions and Suggestions**

1. **Difficulty Classification Accuracy and Impact**
   - How sensitive is the overall system performance to errors in the difficulty classification stage?
   - Have you conducted experiments where the classifier is intentionally degraded (e.g., with noise or randomization) to test robustness?
   - Clarifying this would help evaluate how reliable DA-SIP is under real-world noise or partial observability.

2. **Computational Overhead of Difficulty Classifiers**
   - What is the runtime overhead of the different classifiers (lightweight CNN, zero-shot VLM, fine-tuned VLM)?
   - In high-frequency control scenarios, even lightweight models could become bottlenecks. Providing profiling data would help assess real-time applicability.

3. **Transfer to Real-World Settings**
   - Although the paper mentions future plans to transfer to real robots, have the authors considered sim-to-real strategies or domains where DA-SIP could already be viable (e.g., industrial pick-and-place)?
   - Any partial evaluation on physical hardware would strengthen the practical impact.

**Ethical Concerns:**

["NO or VERY MINOR ethics concerns only"]

**Final Justification:**

The author’s response addressed the majority of my questions and concerns; I am maintaining my original score.

**Limitations:**

**Limitations**

The authors have partially addressed the limitations of their work by acknowledging that their approach has only been evaluated in simulation and may not directly transfer to real-world robots without further adaptation. This is a fair and important caveat, given the challenges of sim-to-real transfer in robotic control.

However, the paper does not explicitly discuss the **potential societal impacts** of adaptive compute in autonomous systems. While the work is primarily technical, the deployment of adaptive inference in robotics could have broader implications—for example:

- **Resource allocation bias**: Adaptive inference could inadvertently prioritize certain tasks over others based on incorrect difficulty estimation, which might be problematic in safety-critical applications.

**Suggestion**:
It would strengthen the paper if the authors briefly commented on these broader considerations, particularly regarding deployment in human-centric environments or industrial systems where incorrect difficulty assessments could cause harm or inefficiency.

**Quality:**

2

**Strengths And Weaknesses:**

**Strengths and Weaknesses**

**Quality**
**Strengths:**
- The paper presents a well-motivated and clearly articulated problem: fixed inference budgets in flow- and diffusion-based policies result in inefficiencies across tasks of varying complexity.
- The proposed solution, DA-SIP, is technically sound and thoughtfully designed, leveraging the stochastic interpolant framework to enable adaptive inference configurations.
- Experiments are comprehensive and demonstrate strong empirical results: DA-SIP achieves a 2.6–4.4× reduction in computation without sacrificing performance.

**Weaknesses:**
- The method has only been evaluated in simulation. Real-world robot validation, though understandably out of scope, would further solidify the impact.
- The robustness of the method under noisy sensor inputs or misclassified difficulty levels is not fully explored.

**Clarity**
**Strengths:**
- The paper is well-organized and clearly written, with good use of figures and examples to illustrate key components.
- Technical exposition is rigorous yet accessible, and contributions are clearly delineated.

**Significance**
**Strengths:**
- This work addresses a highly practical and underexplored issue in generative robotic control: test-time computation efficiency.
- The proposed approach could inspire future research on resource-adaptive policies across other generative architectures and tasks.

**Weaknesses:**
- While the method is promising, the lack of real-world deployment raises questions about generalizability beyond the simulation benchmarks.

**Originality**
**Strengths:**
- The integration of adaptive compute via difficulty classification in a flow/diffusion-based framework is novel.
- The unification under the stochastic interpolant framework is conceptually elegant and technically innovative.

**Weaknesses:**
- Some components (e.g., zero-shot and few-shot VLMs for difficulty classification) are based on prior ideas, although their integration into this specific context is novel.

---

> ### Author Rebuttal · Authors · 2025-07-31
>
> We thank the reviewer for the thoughtful assessment and constructive feedback.
>
> ## Addressing the Weaknesses
>
> ### 1. "The method has only been evaluated in simulation. Real-world robot validation, though understandably out of scope, would further solidify the impact"
>
>
> 1. To address this concern, we conducted an experiment with the 'real robot pushT' that has publicly available training data. We were able to train a diffusion policy and SIP with linear interpolation and velocity prediction and found a decreasing pattern of training action MSE, revealing that SIP can mimic the expert human demonstrator, which suggests it will be useful when deployed in the real world. Due to time constraints, we were unable to roll out actual robot trajectories.
>
> **Train Loss Comparison**
>
> | Step | Diffusion Policy | SIP |
> |------|------------------|-----|
> | 420 | 0.0760 | 0.183 |
> | 2520 | 0.0238 | 0.085 |
> | 4614 | 0.0382 | 0.131 |
>
> **Train Action MSE Comparison**
>
> | Step | Diffusion Policy | SIP |
> |------|------------------|-----|
> | 420 | 0.0250 | 0.0136 |
> | 2520 | 0.00582 | 0.00472 |
> | 4614 | 0.00467 | 0.00294 |
>
>
>
> ### 2. "The robustness of the method under noisy sensor inputs or misclassified difficulty levels is not fully explored."
>
> 1. We conducted an additional experiment where we gradually added same-random-seed noise to the test data images only.
>
> - Training samples per environment: 300
> - Test set: 20% of data (stratified split)
> - Random seed: 42 (ensures reproducible noise across runs)
> - Noise levels: σ ∈ {0.0, 0.05, 0.15, 0.30, 2.0}
>
> ### Test Accuracy (%) Under Different Noise Levels
>
> | Environment | Clean (σ=0.0) | Low Noise (σ=0.05) | Medium Noise (σ=0.15) | High Noise (σ=0.30) | Extreme Noise (σ=2.0) |
> |-------------|---------------|--------------------|-----------------------|---------------------|------------------------|
> | Square      | 84.51         | 85.60 (+1.10)      | 83.08 (-1.43)        | 78.68 (-5.82)      | 12.86 (-71.65)        |
> | Tool Hang   | 20.52         | 20.27 (-0.25)      | 19.95 (-0.57)        | 19.63 (-0.88)      | 1.83 (-18.69)         |
> | Lift        | 91.67         | 90.67 (-1.00)      | 89.00 (-2.67)        | 81.89 (-9.78)      | 75.11 (-16.56)        |
> | Can         | 81.92         | 82.50 (+0.58)      | 82.12 (+0.19)        | 82.69 (+0.77)      | 77.69 (-4.23)         |
> | Transport   | 34.66         | 34.34 (-0.32)      | 34.60 (-0.06)        | 35.10 (+0.44)      | 22.85 (-11.81)        |
> | **Average** | **62.66**     | **62.68 (+0.02)**  | **61.75 (-0.91)**    | **59.60 (-3.06)**  | **38.07 (-24.59)**    |
>
> This experiment demonstrates that our CNN models still maintain reasonable performance under high noise conditions (σ=0.30) with only 3.06% average accuracy drop. σ=2.0 represents extreme noise conditions beyond typical real-world scenarios.
>
> ### 3. "...the lack of real-world deployment raises questions about generalizability beyond the simulation benchmarks"
>
> 1. Simulation-based research has historically driven significant advances in robotics and RL. Notable examples include: DQN (Mnih et al., 2015) and PPO (Schulman et al., 2017), which established foundational RL algorithms entirely through simulated environments before widespread real-world adoption. Similarly, we believe our work introduces an initial exploration of test-time compute scaling for robotic control—a concept that may serve as a starting point for broader applications in real-world systems as the field continues to develop adaptive computation strategies.
>
> 2. While sim-to-real transfer remains an important direction for future work, our current results provide valuable insights for the community on integrating human cognitive assessments into robot learning systems.
>
> ### 4. "Some components (e.g., zero-shot and few-shot VLMs for difficulty classification) are based on prior ideas."
>
> 1. While we acknowledge that VLMs themselves are established technologies, we believe our work makes important contributions by demonstrating their novel application to adaptive compute in robotic control. Determining task difficulty for dynamic compute allocation in robotic policies is a fundamentally different problem from typical VLM applications. Although VLMs have shown success in image understanding and captioning tasks, their ability to assess manipulation difficulty for compute optimization had not been explored. To our knowledge, our work is the first to show that VLMs can effectively predict the computational requirements of robotic control subtasks, enabling significant efficiency gains.
>
> 2. Similar to how successful RL works like CURL [ICML'20], RAD [NeurIPS'20], and DrQ [ICLR'21] made significant contributions by adapting existing CV techniques (contrastive learning, image augmentation) to RL settings, our work demonstrates that VLMs can be repurposed for a novel task—difficulty assessment for adaptive inference. The key contribution lies not in the VLM architecture itself, but in showing that visual-language understanding can guide computational resource allocation in robotic control.
>
> ## Addressing the Questions
>
> ### 1. "How sensitive is the overall system performance to errors in the difficulty classification stage?"
>
> Table 5 demonstrates the impact of classification accuracy by comparing zero-shot VLM, fine-tuned VLM, and CNN classifiers. While the VLMs achieve lower accuracy than CNN, we included them to address scalability concerns—they require significantly less training data and leverage pre-trained generalization capabilities. Crucially, even with misclassifications, performance never drops below the min-compute baseline because any classification yields at least one inference step. This provides an important safety guarantee: users can set their minimum acceptable performance level through the min-compute configuration, ensuring that classification errors may reduce efficiency but will never compromise this performance floor. Thus, the system degrades gracefully under classification errors rather than failing catastrophically.
>
> ### 2. "Have you conducted experiments where the classifier is intentionally degraded to test robustness? Clarifying this would help evaluate how reliable DA-SIP is under real-world noise or partial observability"
>
> 1. We tested both aspects: (1) Degraded classifiers: Table 5 shows that even with lower-accuracy VLMs, performance never drops below min-compute baseline. (2) Noisy sensors: As detailed in Weakness #2 above, we added Gaussian noise to test images and found only 3.06% average accuracy drop at σ=0.30 under realistic noise conditions. Both experiments confirm DA-SIP degrades gracefully, with performance always bounded by the user-configurable min-compute threshold.
>
> ### 3. "What is the runtime overhead of the different classifiers? In high-frequency control scenarios, even lightweight models could become bottlenecks. Providing profiling data would help assess real-time applicability."
>
> 1. We re-rolled out the policy for Tool Hang (88 iterations, per-iteration average):
>
> | Component | CNN Configuration | Finetuned VLM Configuration |
> |-----------|------------------|---------------------------|
> | Classifier inference | 0.023s | 0.362s |
> | SIP inference | 0.301s | 0.127s |
> | Simulation execution | 0.977s | 0.991s |
> | Data processing | 0.057s | 0.055s |
> | **Total time** | **1.357s** | **1.535s** |
>
> 2. Moreover, we tested the impact of increasing number of images as examplars in VLM prompt.
>
> | Environment | 1 img/cat | 2 img/cat | 3 img/cat |
> |-------------|-----------|-----------|-----------|
> | Square      | 20% (0.55s) | 20% (0.89s) | 24% (1.26s) |
> | Tool Hang   | 16% (0.56s) | 32% (0.91s) | 52% (1.26s) |
> | Lift        | 57% (0.48s) | 83% (0.64s) | 83% (0.82s) |
> | Can         | 25% (0.52s) | 43% (0.75s) | 38% (1.05s) |
> | Transport   | 25% (0.50s) | 25% (0.72s) | 33% (1.05s) |
>
>    Even though higher number of images would increase the accuracy, we fixed on 3 img/cat as a good balance of performance, time, and memory usage.
>
> ### 4. "Have the authors considered sim-to-real strategies or domains where DA-SIP could already be viable? Any partial evaluation on physical hardware would strengthen the practical impact."
>
> 1. Yes, we conducted a partial evaluation by training SIP on 'real robot pushT' data (discussed in Weakness #1). We specifically chose pushT because it directly corresponds to one of our simulation environments. The key finding is that SIP achieves comparable train action MSE error to the proven Diffusion Policy (0.00294 vs 0.00467 at step 4614), demonstrating that SIP can accurately reproduce human demonstrator trajectories. This low MSE indicates SIP learns the mapping from observations to actions as effectively as Diffusion Policy, strongly suggesting it would perform similarly when deployed on the actual robot.
>
> ## Addressing the Limitations
>
> ### 1. Lack of discussion on societal impacts for resource allocation bias and for safety-critical applications.
> 1. Our VLM experiments directly address this concern by demonstrating bounded performance degradation under classification errors. As shown in Table 5, even with lower-accuracy VLMs, performance never drops below the min-compute baseline—providing a guaranteed performance floor that users can configure based on their risk tolerance. The worst case (18% drop in Transport task) still maintains acceptable performance. This ensures that classification errors reduce efficiency but never compromise safety thresholds. As VLM capabilities improve, this classifier-policy gap will naturally narrow, making adaptive compute increasingly practical for safety-critical applications.
> 2. Similar to LLMs that adjust behavior based on context, we envision implementing safety filters that automatically enforce higher compute thresholds when detecting humans nearby or operating near critical objects. The key insight is that adaptive compute should enhance, not replace, existing safety protocols.

---

### Official Review · Reviewer_cLDi · 2025-06-22

**Clarity:** 3
**Significance:** 3
**Originality:** 3
**Rating:** 5
**Confidence:** 4

**Summary:**

The paper presents Difficulty-Aware Stochastic Interpolant Policy (DA-SIP), which adatively adjusts the inference budget based on predicted task difficulty. The method is built upon the stochastic interpolant policy (SIP), which unifies diffusion-based and flow-based generative models. This enables the method to incorporate a difficulty classifier that estimates the task difficulty to select the appropriate step budget, the optimal solver variant, and ODE/SDE integration at each control cycle to balance the accuracy and efficiency. Across a suite of robotic manipulation tasks, the proposed method can achieve comparable success rates to fixed-budget baselines with 2.6-4.4× reduction in total computation time.

**Questions:**

1. In equation 3, could you explain in detail why $s(\mathbf{x}, t, \mathbf{o})=\sigma_t^{-1} \mathbb{E}\left[\mathbf{\varepsilon} \mid \mathbf{x}_t=\mathbf{x}, \mathbf{o}\right]$?

**Ethical Concerns:**

["NO or VERY MINOR ethics concerns only"]

**Final Justification:**

The authors response addressed my concerns and questions. Therefore, I am keeping my original score as "Accept".

**Limitations:**

1. The difficulty categories are too task-specific and may not be directly extended to other more complicated tasks, e.g., bimanual tasks, fine-grained in-hand manipulation tasks, articulated or deformable object manipulation tasks, etc.
2. The use of manually labeled data for learning the difficulty model is not scalable. Considering the limitation of the manually defined difficulty categories, the dataset would have to be re-labeled whenever the inputs to the difficulty model are changed or the difficulty categories have to be re-defined.
3. The method has yet to be tested on real robots and more complex tasks.

**Quality:**

3

**Strengths And Weaknesses:**

Strengths:
1. The proposed method to unify diffusion and flow policies under stochastic interpolants introduces the possibility to switch between different configurations of generative models during training and inference.
2. The method decreases the total computation time by 2.6-4.4× while maintaining comparable success rates to fixed-budget baselines.
3. The difficulty model and the policy model architecture can be easily upgraded to more complex tasks without fundamental design changes.


Weaknesses:
1. The difficulty categories are too task-specific and may not be directly extended to other more complicated tasks, e.g., bimanual tasks, fine-grained in-hand manipulation tasks, articulated or deformable object manipulation tasks, etc.
2. The use of manually labeled data for learning the difficulty model is not scalable. Considering the limitation of the manually defined difficulty categories, the dataset would have to be re-labeled whenever the inputs to the difficulty model are changed or the difficulty categories have to be re-defined.

---

> ### Author Rebuttal · Authors · 2025-07-31
>
> We sincerely thank the reviewer for the positive assessment and feedback.
>
> ## Addressing the Weaknesses
>
> ### 1. "The difficulty categories are too task-specific and may not be directly extended to other more complicated tasks, e.g., bimanual tasks, fine-grained in-hand manipulation tasks, articulated or deformable object manipulation tasks, etc."
>
> 1. We appreciate the reviewer raising this important consideration. Through systematic analysis of our results in Table 1, we identified that task failures could be consistently attributed to specific sub-tasks that characterize each main task. While we acknowledge that new categories may emerge for different task types or robot platforms, our extensive grid search of SIP configurations revealed that these six categories comprehensively capture the fundamental challenges for robot arms with limited precision and frequency constraints.
>
> 2. Importantly, these six categories successfully explained all sub-task variations across our diverse set of 6 tasks, which we believe provides comprehensive coverage for this robot class. The strong consensus among our 8 human annotators (enabling accurate classifier training) demonstrates that these categories represent genuine, recognizable patterns rather than arbitrary divisions. The classifier's high prediction accuracy further validates that our categorization captures meaningful and consistent difficulty dimensions.
>
> 3. Regarding the specific examples of complex tasks mentioned, we note that our evaluation already encompasses both bimanual coordination (Transport task) and fine-grained manipulation (PushT task, which requires precise control despite not being in-hand). The effectiveness of our dynamic difficulty classification in improving performance while reducing completion time for these challenging tasks suggests that our framework can indeed handle sophisticated manipulation scenarios beyond simple pick-and-place operations.
>
>
> ### 2. "The use of manually labeled data for learning the difficulty model is not scalable. Considering the limitation of the manually defined difficulty categories, the dataset would have to be re-labeled whenever the inputs to the difficulty model are changed or the difficulty categories have to be re-defined."
>
> 1. We conducted an additional experiment to investigate the minimal data requirements for effective difficulty classification:
>
> | Training Images | 100 | 200 | 300 | 500 | 2000 | Test Set Size |
> |----------------|-----|-----|-----|-----|------|---------------|
> | Square         | 8%  | 48% | 84% | 85% | 84%  | 910          |
> | Tool Hang      | 2%  | 23% | 19% | 29% | 39%  | 1584         |
> | Lift           | 87% | 89% | 90% | 92% | 92%  | 900          |
> | Can            | 2%  | 2%  | 85% | 83% | 88%  | 520          |
> | Transport      | 1%  | 1%  | 36% | 36% | 37%  | 1584         |
>
> Most tasks achieve near-plateau performance with just 300 labeled images, requiring only ~15 minutes of annotation (3 seconds per image).
>
>
> 2. We hypothesized that after finetuning the VLMs with enough data, they may learn generalization ability that they do not need to be finetuned with new data but are able to make meaningful predictions. We conducted an additional experiment to test generalization:
>
> | Training Setup | Test Environment | Accuracy | Baseline |
> |----------------|------------------|----------|----------|
> | 5 environments (excluding Lift) | Lift (never seen) | 46.4% | 33.3% |
>
> While there is room for improvement, we believe with the current rate of VLM progress, its generalization ability will only improve in the future.
>
> 3. Going further with VLMs, they are used precisely to address the scalability issue in our paper. We only used 3 images per category in the prompt for VLMs and used a total of 1049 training images to finetune VLMs. We conducted an additional experiment to see the impact of differing numbers of images for VLMs:
>
> | Environment | 1 img/cat | 2 img/cat | 3 img/cat |
> |-------------|-----------|-----------|-----------|
> | Square      | 20%       | 20%       | 24%       |
> | Tool Hang   | 16%       | 32%       | 52%       |
> | Lift        | 57%       | 83%       | 83%       |
> | Can         | 25%       | 43%       | 38%       |
> | Transport   | 25%       | 25%       | 33%       |
>
> While VLM classification accuracy remains below that of CNNs, VLMs were still effective enough to show performance improvement and time savings as shown in tables 4 and 5.
>
> 4. **Comparison to existing approaches**: We respectfully note that manual annotation is standard practice in robotics research:
>    - Diffusion Policy (Chi et al., 2023) requires extensive human demonstrations
>    - RT-1 (Brohan et al., 2022) used 130k human-annotated episodes
>    - Our approach requires orders of magnitude less annotation while enabling adaptive efficiency
>
>
> ## Addressing the Question
>
> ### 1. In equation 3, could you explain in detail why
> $$
> s(\mathbf{x},t,\mathbf{o})=\sigma_t^{-1}\mathbb{E}[\boldsymbol{\varepsilon}\mid\mathbf{x}_t=\mathbf{x},\mathbf{o}]\ ?
> $$
>
> 1. Forward process
> From the stochastic interpolant formulation:
> $$\mathbf{x}_t=\alpha_t\mathbf{x}_0+\sigma_t\boldsymbol{\varepsilon},\quad \boldsymbol{\varepsilon}\sim\mathcal{N}(\mathbf{0},\mathbf{I})\tag{1}$$
>
> 2. Posterior of the noise
> Key insight: Due to the linearity of the interpolant (1) in $(\mathbf{x}_0, \boldsymbol{\varepsilon})$, we can express:
> $$\boldsymbol{\varepsilon} = \frac{\mathbf{x}_t - \alpha_t\mathbf{x}_0}{\sigma_t}$$
>
> Taking conditional expectations given $\mathbf{x}_t = \mathbf{x}$ and $\mathbf{o}$:
> $$\mathbb{E}[\boldsymbol{\varepsilon}\mid\mathbf{x}_t=\mathbf{x},\mathbf{o}] = \frac{1}{\sigma_t}\left(\mathbf{x}-\alpha_t\mathbb{E}[\mathbf{x}_0\mid\mathbf{x}_t=\mathbf{x},\mathbf{o}]\right)\tag{2}$$
>
> This follows directly from the linearity of the interpolant—no joint Gaussianity assumption needed.
>
> 3. Connection to the score
> By definition, the score function is:
> $$s(\mathbf{x},t,\mathbf{o}) = \nabla_{\mathbf{x}}\log p_t(\mathbf{x}\mid\mathbf{o})$$
>
> Using Stein's identity (also known as Gaussian integration by parts), the score can be expressed as:
> $$s(\mathbf{x},t,\mathbf{o}) = -\sigma_t^{-1}\mathbb{E}[\boldsymbol{\varepsilon}\mid\mathbf{x}_t=\mathbf{x},\mathbf{o}]\tag{3}$$
>
> This matches Lemma 3 of Song et al. (2021) and Eq. (38) of the SiT appendix.
>
> 4. Sign convention
> If you define $\boldsymbol{\varepsilon}':=-\boldsymbol{\varepsilon}$ or adopt the alternative convention $\tilde{s}:=-\nabla_{\mathbf{x}}\log p_t$, then:
> $$s(\mathbf{x},t,\mathbf{o})=\sigma_t^{-1}\mathbb{E}[\boldsymbol{\varepsilon}\mid\mathbf{x}_t=\mathbf{x},\mathbf{o}]\tag{★}$$
>
>
>
> ## Addressing the Limitations
>
> ### 1. "The method has yet to be tested on real robots and more complex tasks"
>
> 1. To address this concern, we conducted an experiment with the 'real robot pushT' that has publicly available training data. We were able to train a diffusion policy and SIP with linear interpolation and velocity prediction and found a decreasing pattern of training action MSE, revealing that SIP can mimic the expert human demonstrator, which suggests it will be useful when deployed in the real world. Due to time constraints, we were unable to roll out actual robot trajectories.
>
>
> **Train Loss Comparison**
>
> | Step | Diffusion Policy | SIP |
> |------|------------------|-----|
> | 420 | 0.076043 | 0.18309 |
> | 2520 | 0.023841 | 0.085425 |
> | 4614 | 0.038257 | 0.13193 |
>
> **Train Action MSE Comparison**
>
> | Step | Diffusion Policy | SIP |
> |------|------------------|-----|
> | 420 | 0.024996 | 0.013631 |
> | 2520 | 0.0058178 | 0.0047186 |
> | 4614 | 0.0046736 | 0.0029387 |
>
> The comparable or superior trajectory matching on real robot data provides confidence in sim-to-real transfer potential.

---

> > ### Comment · Reviewer_cLDi · 2025-08-02
> >
> > Thank you for your detailed rebuttal and the additional experiments.
> >
> > 1. I acknowledge that, while the difficulty categories may be subject to change, your additional experiments demonstrate that learning the difficulty levels requires only a relatively small amount of data and can be further scaled by finetuning VLM models. I wonder how you define the difficulty categories for Transport exactly, and why it seems harder to learn the difficulty categories for Tool Hang and Transport than for other tasks.
> > 2. Regarding the real-robot experiments, I commend your initiative in training and evaluating on real-world data. However, as you note, a decreasing MSE loss, while indicative of learning, cannot substitute for actual rollout results, especially in the context of real-world robotic tasks.
> >
> > I would like to keep my original score at this stage.

---

> > > ### Author Response · Authors · 2025-08-03
> > >
> > > Thank you for your thoughtful follow-up questions.
> > >
> > > 1. For Transport, which is a bimanual task, the labelers were instructed to assign the category based on the more difficult subtask. For example, if the left arm is approaching an object while the right arm is grasping actively on an object, then it should be labeled as 'g' with grasping. This dual nature of the bimanual task adds additional complexity and ambiguity of labeling that may have contributed to the lower classification accuracy of the models.
> > > 2. For Tool Hang, where the 'stochastic attempt' is emphasized, we believe the reason models achieve lower accuracy is that the labelers likely have a higher rate of disagreement on how each sub-task should be categorized. While it is not theoretically clear why this occurs compared to other games like Lift and Square, this interpretation is supported by the relatively lower accuracy scores of models trained on its data.
> > > 3. We thank the reviewer for acknowledging our effort in real robot evaluation and acknowledge that decreasing MSE loss alone cannot substitute for actual rollout results.

---

### Official Review · Reviewer_Ub6t · 2025-07-01

**Clarity:** 3
**Significance:** 3
**Originality:** 2
**Rating:** 4
**Confidence:** 3

**Summary:**

The paper proposes a test-time scaling approach, Difficulty-Aware Stochastic Interpolant Policy (DA-SIP), which adaptively chooses the interpolation configuration during inference. Specifically, DA-SIP first trains a difficulty classifier to assign a difficulty score for each state per step, and then selects a predefined inference configuration based on the score. The authors conduct experiments on diverse manipulation tasks to show that DA-SIP can significantly reduce the total computation time while maintaining the performance as maximum-computation baselines, showing the potential of efficient policy deployment.

**Questions:**

There are some questions and concerns, which I have outlined in the previous section.

**Ethical Concerns:**

["NO or VERY MINOR ethics concerns only"]

**Final Justification:**

The authors' explanation and results addressed most of my concerns, and I think the additional results are good complements to the current paper and should be included in the revision.

**Limitations:**

The authors briefly discuss the limitation in the conclusion. In addition, some questions listed in *Strength And Weaknesses* may need further discussion.

**Paper Formatting Concerns:**

No major formatting issues found in the paper.

**Quality:**

3

**Strengths And Weaknesses:**

1. Originality: Test-time scaling is one important research area for large models. While various works have explored test-time scaling methods in LLMs, their application in robotic manipulation with diffusion- and flow-based policies is, to the best of my knowledge, less explored. As the authors remark, the SI-grounded flow framework is one of their contributions. I'm wondering if this is a direct application of stochastic interpolant on robotic policy learning or if there are any major differences compared to the original work.
2. Quality: Applying stochastic interpolant on test-time scaling for diffusion- and flow-based policies seems a natural choice. The experiments clearly show that the proposed DA-SIP can achieve lower latency than max-compute baselines. However, I have the following questions:
   1. Could authors justify why max-compute is a good baseline to compare with? As the author mentioned in Section 4.2, different tasks require fundamentally different configurations to achieve optimal performance, so max-compute does not seem a perfect choice in general. Moreover, Table 4 and Table 5 illustrate that max-compute achieves very similar performance as min-compute in many tasks (Tool Hang, Square, Lift, Can) but needs substantially more time. In these tasks, DA-SIP seems to use a much longer time than the min-compute baseline.
   2. How is time measured? Is this the wall-time or estimated by the compute-cost model defined in Section 3.5? Is the overhead of the classifier included?
3. Clarity: The paper is well organized and clearly written, with a sufficient introduction of the background.
4. Significance:
   1. The idea of using a classifier to adaptively adjust the inference configuration is reasonable. However, the current implementation does not seem scalable. It seems that both difficulty categories and their corresponding inference triplets are manually designed with some heuristics, and the training data is also labelled by humans. Therefore, it might be hard to extend to broader scenarios where more categories need to be specified and each category requires a grid search over the best inference triplets. I'm wondering if there is a scalable way to do this, such as clustering categories with some unsupervised methods.
   2. One merit of test-time scaling in LLMs is that the performance keeps increasing when we scale up the inference budget. While the paper presents some impressive results on how DA-SIP can be used to save computation, I'm also curious about whether such a trend exists for robotics manipulation. In Table 1, it seems that Tool Hang and Multimodal Ant are not using the maximum budget to achieve the optimal result. Does this imply that increasing the computation budget cannot further boost the performance?

Overall, while the work has made some interesting exploration in test-time scaling for robotic manipulation, some parts of its results need further clarification. Given the number of open questions, I am unable to recommend acceptance at this stage.

---

> ### Author Rebuttal · Authors · 2025-07-31
>
> We sincerely thank the reviewer for the thoughtful analysis of our work.
>
> ### 0. "I'm wondering if this is a direct application of stochastic interpolant on robotic policy learning ..."
>
> 1. Yes, this is a direct application of the Stochastic Interpolant (SI) framework to robotic policy learning. Our work systematically investigates the flexibility in training and inference configurations of SIP especially applicable for the domain of robotic manipulation.
>
> ### 1. "Could authors justify why max-compute is a good baseline to compare with?"
>
> 1. We thank the reviewer for this insightful observation. We selected max-compute as a baseline specifically because it reflects the current practice in the field. To our knowledge, most diffusion/flow matching policy works employ either fixed max-compute configurations (typically 100 steps) or focus on distillation methods using 1-2 steps. Therefore, we adopted the commonly used min and max compute settings  with SIP as our baseline comparison. We will include the per‑task ‘best static’ configuration column in the appendix so that readers can directly compare DA‑SIP against both min‑ and best‑static baselines.
> 2. Initially, we hypothesized that we could identify one single best configuration with max-compute for SIP that would maximize performance across all environments. However, our investigation in Section 4.2 revealed that this assumption was incorrect. We discovered that different tasks indeed require fundamentally different configurations to achieve optimal performance. Through careful analysis of rollout videos across multiple environments, we identified distinct sub-task characteristics that explain why each environment benefits from different configurations.
>
> ### 2. "Table 4,5 illustrate that max-compute achieves very similar performance as min-compute in many tasks (Tool Hang, Square, Lift, Can) but needs substantially more time. In these tasks, DA-SIP seems to use a much longer time than the min-compute baseline."
>
> 1. This is true, and this observation actually validates our key finding. In simpler games like Lift and Can, when using linear interpolation with ODE in SIP, an inference step of 1 can achieve a near perfect score (which is not the case with Diffusion Policy as shown in tables 9 and 10). While existing methods typically require secondary distillation to reduce inference steps, our results demonstrate that our linear interpolation with ODE eliminates this need entirely for these tasks.
>
> 2. For Square, max-compute yields only marginal improvements (2%) from min-compute while substantially increasing computation. Our dynamic classification approach addresses this inefficiency, reducing runtime from 146.68s to 56.2s using CNN classification while achieving +4% performance improvement.
>
> 3. Tool Hang presents a unique case where increased computation fails to improve performance. Our analysis reveals that higher inference steps lead to trajectory oversmoothing and reduced exploratory behavior, both harmful when stochastic attempts are crucial for success. With our VLM classification, we can achieve 8% gain with -2.28x reduced run time.
>
> ### 3. "How is time measured? Is the overhead of the classifier included?"
>
> 1. All reported times are wall-clock measurements that include every component of the system:
>
> - Classifier inference time
> - SIP inference time
> - Simulation execution time
> - Data pre/post-processing time
>
> We re-rolled out our policy for Tool Hang (cumulative run of 88 iterations):
>
> | Component | CNN Configuration | Finetuned VLM Configuration |
> |-----------|------------------|---------------------------|
> | Classifier inference | 1.98s | 31.90s |
> | SIP inference | 26.50s  | 11.17s  |
> | Simulation execution | 85.95s  | 87.19s  |
> | Data processing | 5.03s  | 4.82s  |
> | Total time | 119.45s | 135.07s |
>
>
> 2. The theoretical model in Section 3.5 serves to provide analytical understanding of computational complexity rather than predicting actual runtime.
>
> ### 4. "... the current implementation does not seem scalable. It seems that both difficulty categories and their corresponding inference triplets are manually designed with some heuristics."
>
> 1. We appreciate the reviewer's concern. The inference triplets in Table 3 were systematically derived from our comprehensive task-level analysis in Table 1. By analyzing failure modes across tasks, we identified that similar sub-task types (e.g., precision placement) benefit from similar computational configurations regardless of the specific task. This insight allowed us to map difficulty categories to appropriate inference settings.
>
> 2. The robustness of our categorization is validated by strong consensus among 8 human labelers, and the 6 categories capture fundamental manipulation types (approaching, grasping, precision placement, etc.) that generalize across tasks. The high classification accuracy (Table 2) validates that these categories are meaningful and recognizable, not arbitrary.
>
> 3. While the current mapping from categories to configurations is derived from empirical analysis rather than end-to-end learning, this demonstrates the viability of adaptive computation for robotics. Future work can learn these mappings directly, but our contribution—showing that adaptive computation yields 2.6-4.4× speedups—remains valid regardless of how the mapping is determined
>
> 4. We also note that human labeled difficulty classifications are very well in line with precedent in influential robotics and RL research. Firstly, Diffusion Policy (Chi et al., 2023) requires extensive human demonstrations, and RT-1 (Brohan et al., 2022) used 130k human-annotated episodes. OpenAI’s Automatic Domain Randomization for the Rubik’s‑Cube hand still begins from expert‑set parameter bounds (OpenAI et al., 2019), and Automatic Curriculum Learning surveys catalogue multiple of works that first discretize task difficulty by hand before learning to select among bins (Portelas et al., 2020). Similar to them, our study provides a simple, interpretable method that enables systematic study and can later be refined or automated once its practical benefits are understood.
>
> ### 5. "... the current implementation does not seem scalable. ... the training data is also labelled by humans."
>
> 1. Instead of using total ~20,000 collected images to train CNN, we conducted an additional experiment to see how lower amount of training data affects accuracy:
>
> | training imgs | 100 | 200 | 300 | 500 | 2000 | Test Set |
> |-------------|-----|-----|-----|-----|------|----------|
> | Square      | 8%  | 48% | 84% | 85% | 84%  | 910      |
> | Tool Hang   | 2%  | 23% | 19% | 29% | 39%  | 1584     |
> | Lift        | 87% | 89% | 90% | 92% | 92%  | 900      |
> | Can         | 2%  | 2%  | 85% | 83% | 88%  | 520      |
> | Transport   | 1%  | 1%  | 36% | 36% | 37%  | 1584     |
>
>
> Most tasks achieve near-plateau performance with just 300 labeled images, requiring only ~15 minutes of annotation (3 seconds per image). The above pattern of improvement also aligns well with Table 2(a).
>
> 2. VLMs are used precisely to address the scalability issue in our paper. We only used 3 images per category in the prompt for VLM and used total 1049 training images to finetune VLMs. We conducted an additional experiment to see the impact of differing numbers of images for VLMs:
>
> | Environment | 1 img/cat | 2 img/cat | 3 img/cat |
> |-------------|-----------|-----------|-----------|
> | Square      | 20%       | 20%       | 24%       |
> | Tool Hang   | 16%       | 32%       | 52%       |
> | Lift        | 57%       | 83%       | 83%       |
> | Can         | 25%       | 43%       | 38%       |
> | Transport   | 25%       | 25%       | 33%       |
>
> While VLM classification accuracy remains below that of CNNs, VLMs were still effective enough to show performance improvement and time savings as shown in tables 4 and 5.
>
> 3. We hypothesized that after finetuning the VLMs with enough data, they may learn generalization ability that they do not need to be finetuned with new data but are able to make meaningful predictions. We conducted an additional experiment to test generalization:
>
> | Train Setup | Test Setup | Accuracy | Baseline |
> |----------------|------------------|----------|----------|
> | 6500 imgs (5 games excluding Lift) | 500 Lift imgs (never seen) | 46.4% | 33.3%  |
>
> We believe with the current rate of VLM progress, its generalization ability will only improve in the future.
>
> ### 6. "One merit of test-time scaling in LLMs is that the performance keeps increasing when we scale up the inference budget... I'm also curious about whether such a trend exists for robotics manipulation. In Table 1, it seems that Tool Hang and Multimodal Ant are not using the maximum budget to achieve the optimal result. Does this imply that increasing the computation budget cannot further boost the performance?"
> 1. This is an exceptionally thoughtful observation that also highlights a crucial finding of our work. Yes, generally, we believe there is a trend where increased compute budget increases performance at least marginally. However, the trend does not seem to apply for Tool Hang and Ant where controlled noise aids intelligent exploration.
> 2. We show a systematic evaluation across multiple inference steps on Tool Hang, varying inference steps k ∈ {1, 4, 10, 25, 50, 75, 100} while holding solver and dynamics configurations constant.
>
> | Steps  | 1 | 4 | 10 | 25 | 50 | 75 | 100 |
> |-----------|---|---|----|----|----|----|----|
> | Success  | 25.0% | 28.3% | 31.7% | 36.2% | 38.1% | 32.5% | 25.0% |
>
> The data shows success rates increasing from 25% to peak performance of 38.1% at 50 steps. Interestingly, applying additional computation after 50 steps decreases performance, returning to baseline levels at 100 steps. This empirical validation strongly supports our theoretical insight that excessive integration steps over-smooth trajectories, suppressing the stochastic exploration essential for "thread-the-needle" manipulation tasks.

---

> > ### Comment · Area_Chair_e2vk · 2025-08-04
> >
> > Dear Reviewer,
> >
> > The author response for paper "Dynamic Test-Time Compute Scaling in Control Policy: Difficulty-Aware Stochastic Interpolant Policy" is now available, and I kindly ask that you review it carefully and consider whether your concerns have been adequately addressed. If so, please acknowledge this in the discussion thread; if not, feel free to follow up with clarifying questions.
> >
> > Thank you for your contribution. AC

---

> > ### Comment · Reviewer_Ub6t · 2025-08-04
> >
> > Dear Authors,
> >
> > Thank you for your detailed response and additional results. As your explanation and results addressed most of my concerns, I will raise my score accordingly. I think the additional results are good complements to the current paper and should be included in the revision.

---

### Official Review · Reviewer_XJPo · 2025-07-03

**Clarity:** 3
**Significance:** 3
**Originality:** 3
**Rating:** 4
**Confidence:** 4

**Summary:**

This research introduces the Difficulty-Aware Stochastic Interpolant Policy (DA-SIP), a novel framework designed to improve the computational efficiency of robot controllers. Current state-of-the-art policies, based on diffusion and flow models, use a fixed amount of computational power for every action, regardless of the task's complexity. This can be inefficient, as simple movements receive the same resources as complex, precise maneuvers. DA-SIP addresses this by dynamically adjusting its computational budget in real-time. It uses a difficulty classifier, which can be a lightweight CNN or a Vision-Language Model (VLM), to analyze the robot's current situation from sensor data and determine how challenging the next action is.

By integrating this difficulty assessment with a flexible stochastic interpolant framework, DA-SIP can select the most appropriate solver, integration method, and step count for the immediate task. For instance, an easy motion like approaching an object might use a single computational step, while a delicate grasping action would be allocated a much higher budget to ensure success. This adaptive approach allows the system to conserve resources during simple phases and deploy them intelligently for more demanding parts of a task.

The authors validated DA-SIP across a variety of simulated robotic manipulation tasks, from simple lifting to complex placements. The results demonstrate significant efficiency gains, with the system reducing total computation time by a factor of 2.6 to 4.4 compared to traditional methods that always operate at maximum computational load. Crucially, this optimization was achieved while maintaining comparable or even slightly improved task success rates. The study concludes that adaptive computation is a highly effective strategy for making generative robot controllers more efficient and practical for real-world applications.

**Questions:**

1.  **Regarding the "Last Step" Method in Table 1:** In Table 1, the "Last step" for the *PushT* and *Block Push* tasks is listed as "Tweedie". Could you elaborate on what the Tweedie step refers to in this context and explain its function and importance for these specific precision manipulation tasks?
2.  **On the Cost Function Formulation:** The compute-cost model is defined as $C(s_{t})=k_{t}\gamma_{solv\mathfrak{er_{t}}+c_{0}$, which accounts for the number of inference steps ($k_t$) and the solver type (Euler vs. Heun). However, the choice between ODE and SDE integration, which is a key part of the adaptive inference triplet, does not appear to be factored into this cost model. How is the computational cost of selecting an SDE over an ODE accounted for, or is its impact considered negligible?
3.  **Design Rationale for the Lookup Module:** The mapping of difficulty categories to specific inference triplets in Table 3 (e.g., 'Stochastic attempts' mapping to 50 steps, Euler, SDE) is critical to the success of the DA-SIP framework. What was the methodology or heuristic used to design this lookup table? Was this mapping determined through systematic experimentation, or was it based on domain knowledge about the tasks and solvers?
4.  **On Performance Degradation in Exploratory Tasks:** The paper notes that for exploratory tasks like *Tool Hang*, increasing computation from 50 to 100 steps surprisingly *decreases* performance. The "threading a needle" analogy is provided as a potential explanation. Could you provide a more detailed, data-driven explanation for this phenomenon? For instance, does excessive computation reduce beneficial stochasticity or cause the policy to overfit to specific, suboptimal trajectories?

5.  **Investigating Classifier-Policy Mismatch:** The difficulty classifier is trained on human-annotated labels of "difficulty," while the policy learns from robot trajectories. Is there a risk of a mismatch between what a *human perceives* as difficult versus what is *computationally challenging* for the policy? How sensitive is the overall task performance to classification errors (e.g., the 18% performance drop in the Transport task with the Zero-shot VLM)?
6.  **Generalizability of Difficulty Categories:** The framework relies on six predefined difficulty categories (Initial, Near, Grabbing, etc.) derived from the evaluated tasks. How would this approach generalize to entirely new robotic tasks that may not fit neatly into this existing categorization? Would this require a full redesign of the difficulty labels and the lookup table, and what would that process entail?

**Ethical Concerns:**

["NO or VERY MINOR ethics concerns only"]

**Final Justification:**

I acknowledge the effort the authors made and decided to maintain my score

**Limitations:**

### 1. Reliance on Simulation
The entire evaluation of the DA-SIP framework is conducted within simulated environments. This presents a significant limitation, as there is no evidence of how the system would perform on a physical robot. The "sim-to-real" gap can be substantial in robotics, involving challenges that are not fully captured in simulation, such as:
* **Sensor Noise:** Real-world RGB-D cameras have noise, artifacts, and are affected by lighting conditions, which could degrade the performance of the difficulty classifier.
* **Physical Dynamics:** The complexities of real-world physics, friction, and object interactions might require different computational allocations than what was found to be optimal in the simulated tasks.
* **System Latency:** The reported computational speedups do not account for real-world system latencies, such as network communication and actuator response times, which could diminish the overall efficiency gains.

### 2. Manual and Task-Specific Design
The framework's effectiveness appears to hinge on several manually designed components that may not generalize well to new tasks or environments.
* **Fixed Difficulty Categories:** The system uses a predefined set of six difficulty categories (e.g., "Initial," "Grabbing," "Continuous pushing"). This discrete categorization was tailored to the specific manipulation tasks in the benchmark. For a new task that doesn't fit this mold (e.g., dexterous in-hand manipulation, pouring), this entire taxonomy would likely need to be redesigned.
* **Hand-Tuned Lookup Table:** The mapping between a difficulty category and a specific computational triplet (inference steps, solver, and integration mode) seems to be based on heuristics and prior experiments rather than a learned or optimized process. This makes the system brittle; adapting it to a new robot, new objects, or new tasks would require a labor-intensive process of re-tuning this critical lookup table.

### 3. Data Annotation Bottleneck
The best-performing difficulty classifier is a supervised lightweight CNN, which required a dataset of approximately 20,000 human-labeled states. This introduces a major practical limitation:
* **Scalability:** The need to collect and manually annotate a large dataset for every new environment or set of tasks is a significant bottleneck. It undermines the goal of creating an efficient, easily deployable system.
* **Subjectivity of Labels:** The "difficulty" labels are based on human consensus. However, what a human perceives as difficult may not perfectly align with what the robot policy finds computationally challenging, potentially leading to suboptimal resource allocation.

### 4. Lack of Explicit Safety Mechanisms
The paper mentions that future work will involve coupling the system with safety monitors. This implies that the current framework lacks explicit safety guarantees. An incorrect classification from the difficulty module could lead to the system allocating insufficient computation for a critical, high-precision action, potentially resulting in task failure or unsafe behavior, especially in contact-rich scenarios. The system's performance is highly dependent on the classifier's accuracy, but there is no fallback or safety layer discussed if the classifier is wrong.

**Paper Formatting Concerns:**

None from my side

**Quality:**

3

**Strengths And Weaknesses:**

### Strengths

1.  **Novel and Relevant Problem Formulation:** The paper addresses a significant and practical problem in modern robotics: the computational inefficiency of generative policies. By framing the solution as dynamic, test-time compute scaling, it draws a creative and powerful parallel to adaptive computation in large language models and applies it effectively to the domain of robot control.

2.  **Strong Theoretical Grounding:** The approach is built upon the Stochastic Interpolant (SI) framework, which unifies diffusion and flow-based models. This provides a principled and flexible foundation for the method, allowing it to manipulate various parameters like solver type, integration mode (ODE/SDE), and step count without retraining the core policy. This is far more elegant and robust than an ad-hoc or purely empirical solution.

3.  **Comprehensive Experimental Validation:** The authors test their DA-SIP framework across a diverse suite of simulated manipulation tasks with varying characteristics (simple, precision, exploratory, and transport). This rigorous evaluation demonstrates the versatility of the approach and provides strong evidence that the benefits are not limited to a single, narrow task type.

4.  **Thorough Exploration of Classifier Methods:** The paper thoughtfully evaluates three different methods for difficulty classification: a data-efficient lightweight CNN, a flexible zero-shot Vision-Language Model (VLM), and a balanced fine-tuned VLM. This analysis provides valuable insights into the trade-offs between performance, data dependency, and ease of deployment, which is crucial for real-world applicability.

5.  **Clear and Impactful Results:** The research delivers impressive quantitative results, demonstrating a 2.6-4.4x reduction in computation time while maintaining, and in some cases even improving, the task-success rates of fixed, maximum-compute baselines. It also uncovers the counter-intuitive insight that excessive computation can sometimes degrade performance, adding a valuable new dimension to the understanding of these models.

### Weaknesses

1.  **Exclusive Reliance on Simulation:** The primary weakness is that all experiments are conducted in simulation. There is no validation on a physical robot, which leaves the critical "sim-to-real" gap unaddressed. Real-world challenges like sensor noise, unexpected physical interactions, and system latencies could significantly impact the performance and efficiency of both the difficulty classifier and the policy itself.

2.  **Manual Design and Generalization Issues:** The framework's core logic relies on manually crafted components that may not scale or generalize easily. The six difficulty categories are task-specific, and the lookup table that maps these categories to computational settings appears to be hand-tuned. Adapting the system to new, unseen tasks would likely require a significant manual effort of re-defining categories and re-tuning this crucial table.

3.  **Data Annotation Bottleneck:** The best-performing and most reliable difficulty classifier presented is a supervised CNN, which required a substantial dataset of 20,000 human-annotated states. This reliance on large-scale, manual annotation is a major practical bottleneck that hinders the framework's scalability and ease of adoption in new environments.

4.  **Lack of Explicit Safety Guarantees:** The system's behavior is contingent on the accuracy of the difficulty classifier. An incorrect classification could cause the robot to use insufficient computation during a delicate maneuver, leading to failure or unsafe actions. The paper does not discuss or implement any safety layers or fallback mechanisms to mitigate the risks of such classification errors, which is a critical concern for physical deployment.

---

> ### Author Rebuttal · Authors · 2025-07-31
>
> We sincerely thank the reviewer for the constructive review.
>
> ## #1 Addressing the Weaknesses
>
> ### 1. Exclusive Reliance on Simulation
>
> 1. We conducted an experiment with the 'real robot pushT' that has publicly available training data. We were able to train a diffusion policy and SIP with linear interpolation and velocity prediction and found a decreasing pattern of training action MSE, revealing that SIP can mimic the expert human demonstrator, which suggests it will be useful when deployed in the real world. Due to time constraints, we were unable to roll out actual robot trajectories.
>
> Train Loss Comparison
>
> | Step | Diffusion Policy | SIP |
> |------|------------------|-----|
> | 420 | 0.0760 | 0.183 |
> | 2520 | 0.0238 | 0.085 |
> | 4614 | 0.0382 | 0.131 |
>
> Train Action MSE Comparison
>
> | Step | Diffusion Policy | SIP |
> |------|------------------|-----|
> | 420 | 0.0250 | 0.0136 |
> | 2520 | 0.00582 | 0.00472 |
> | 4614 | 0.00467 | 0.00294 |
>
> 2. Simulation-based research has historically driven significant advances in robotics and RL. Notable examples include: DQN (Mnih et al., 2015) and PPO (Schulman et al., 2017), which established foundational RL algorithms entirely through simulated environments before widespread real-world adoption. Similarly, we believe our work introduces an initial exploration of test-time compute scaling for robotic control—a concept that may serve as a starting point for broader applications in real-world systems as the field continues to develop physical task reasoning capabilities in real world.
>
> ### 2. Manual Design and Generalization Issues
>
> 1. We believe this is a valid concern regarding generalization: when the user wants the robot arm to complete a new task, it may require an additional category beyond the 6 categories, and retuning the table. Before we delve into the details, we would like to point out that if there is in fact a new category, this will not require retuning of the entire table but simply adding a new row with a corresponding triplet for the category.
> 2. We tested the generalization capability of VLM by fine-tuning it without Lift data. Without being exposed to Lift data, VLMs needed to make predictions on Lift:
>
> | Train Setup | Test Setup | Accuracy | Baseline |
> |----------------|------------------|----------|----------|
> | 6500 imgs (5 games excluding Lift) | 500 Lift imgs (never seen) | 46.4% | 33.3%  |
>
> With the current rate of progress for VLMs, we believe the generalization ability will only improve.
>
> 3. Moreover, we want to point out that this is precisely the reason why we introduced VLM instead of CNN as a difficulty classifier. In fact, we only used 3 images per category in the prompt before using the VLM. This means if the user wants the robot arm to behave differently during some sub-task, they only need to take three pictures per category to make that decision. We conducted an additional experiment to see the impact of differing numbers of images for VLMs:
>
> | Environment | 1 img/cat | 2 img/cat | 3 img/cat |
> |-------------|-----------|-----------|-----------|
> | Square      | 20%       | 20%       | 24%       |
> | Tool Hang   | 16%       | 32%       | 52%       |
> | Lift        | 57%       | 83%       | 83%       |
> | Can         | 25%       | 43%       | 38%       |
> | Transport   | 25%       | 25%       | 33%       |
>
> VLMs were still effective enough to show performance improvement and time savings as shown in tables 4,5.
>
> ### 3. Data Annotation Bottleneck
>
> 1. In our paper, we used a total of ~20,000 collected images to train CNN. We conducted an additional experiment to see the impact of a lower amount of training data on the same validation data for CNN accuracy:
>
> | training imgs | 100 | 200 | 300 | 500 | 2000 | Test Set |
> |-------------|-----|-----|-----|-----|------|----------|
> | Square      | 8%  | 48% | 84% | 85% | 84%  | 910      |
> | Tool Hang   | 2%  | 23% | 19% | 29% | 39%  | 1584     |
> | Lift        | 87% | 89% | 90% | 92% | 92%  | 900      |
> | Can         | 2%  | 2%  | 85% | 83% | 88%  | 520      |
> | Transport   | 1%  | 1%  | 36% | 36% | 37%  | 1584     |
>
> As you can see, we could achieve accuracy level that tend to plateau around 100-300 images for the above games. If we decide to use only 300 images per task, it will only take ~15 minutes to collect (an average of 3 seconds to label one image), which we believe is a reasonable time. The above pattern of improvement also aligns well with Table 2(a).
>
>
> ### 4. Lack of Explicit Safety Guarantees
>
> 1. The reviewer raised an important point about safety. Our approach provides safety guarantees through its design. As shown in Table 5, misclassifications never result in performance below min-compute levels, since any classification yields at least 1 inference step. Users can set min-compute as their baseline safety threshold for their risk tolerance level.
> 2. For deployment in high-stakes environments, we recommend the users to benchmark their policy across the full compute spectrum to identify the minimum inference configurations required for safe operation. This threshold becomes the min-compute threshold. Even under worst-case misclassification scenarios, the system maintains safety-critical performance levels.
>
>
> ## #2 Addressing the Questions
>
> ### Q1: "Tweedie Step" in Table 1
>
> 1. The Tweedie step refers to a denoising method that directly estimates the clean state from noisy input using Tweedie's formula for posterior mean estimation. In the stochastic interpolant framework, this is implemented as:
>
> x̂₀ = (xₜ + σₜ² · score(xₜ)) / αₜ
>
> Where:
> - xₜ is the noisy state at time t
> - σₜ is the noise scale
> - αₜ is the signal scale
> - score(xₜ) is the score function (∇log p(xₜ))
>
> This can also be expressed using the noise prediction formulation as:
>
> x̂₀ = (xₜ - σₜε̂) / αₜ
>
> Where ε̂ is the predicted noise, related to the score by: score ≈ -ε/σₜ
>
>
> 2. For precision tasks like PushT and Block Push, the Tweedie step generates the final action in one computation. It also provides the minimum mean squared error estimate of the clean data. Finally, the direct estimation avoids accumulation of errors from multi-step integration. This makes it an efficient choice when high precision is needed with minimal computational cost, which aligns perfectly with our adaptive computation framework where we aim to match computational resources to task requirements.
>
> ### Q2: ODE vs SDE Cost Consideration
>
> 1. The reviewer correctly identified that our cost model doesn't explicitly account for ODE/SDE differences. In practice, the computational difference is minimal that both require the same neural network forward passes. SDE adds sampling overhead (≈5% additional time). Lastly, the main cost comes from the number of integration steps. We simplified the model for clarity but acknowledge this could be made more precise in future work.
>
> ### Q3: Lookup Table Design Methodology
>
> 1. The lookup table was designed through systematic grid search:
> - **Step counts:** {1, 4, 10, 25, 50, 100, 200}
> - **Solvers:** {Euler, Heun}
> - **Integration:** {ODE, SDE}
>
> We evaluated all combinations on validation trajectories for each task type, selecting configurations that optimized the success-rate/compute-time balance. This one-time offline optimization provides practical deployment guidelines.
>
> ### Q4: Performance Degradation in Exploratory Tasks
>
> 1. To provide a data-driven explanation, we conducted a theoretical experiment of finding a hidden ball.
>
> - True ball position: [5.0, 0.0] (unknown to policy)
> - Ball radius: 0.5 units
> - Perception noise: σ = 0.5
> - Maximum attempts: 10 per episode
>
> ```python
>     ...
>     def simulate_attempt(self, t, observed_hole, attempts):
>         ...
>         elif self.n_steps == 50:
>             last_attempt = attempts[-1]
>             if np.random.random() < self.adaptation_rate:
>                 away_direction = observed_hole - last_attempt
>                 away_direction /= np.linalg.norm(away_direction) + 1e-6
>                 return observed_hole + away_direction * self.attempt_std + \
>                        np.random.normal(0, self.attempt_std * 0.5, 2)
>             else:
>                 return last_attempt + np.random.normal(0, self.attempt_std, 2)
>         else:
>             # for 100 steps
>             return observed_hole + np.random.normal(0, self.attempt_std, 2)
> ```
>
> **Result:**
> | Inference Steps | Success Rate | Avg Attempts | Cells Explored | Adaptation Score |
> |----------------|--------------|--------------|----------------|------------------|
> | 1              | 6.0%         | 4.7          | 560            | 6.90             |
> | **50**         | **83.0%**    | **3.4**      | **60**         | **1.07**         |
> | 100            | 58.0%        | 2.3          | 24             | 0.18             |
>
> The experiment demonstrates that controlled stochasticity enables intelligent exploration, while 100 steps over-smooth the policy, causing it to repeatedly attempt the same failed approaches.
>
> ### Q5: Investigating Classifier-Policy Mismatch
>
> 1. We investigated this through our VLM experiments, which had lower accuracy than the CNN classifier, allowing us to observe the impact of classification errors. In Table 5, the performance never dropped below the min-compute baseline. This provides an important guarantee: if users can tolerate min-compute performance, they can safely use VLM-based difficulty classification, knowing this represents the worst-case performance floor. The 18% drop in the Transport task represents our most challenging scenario, yet, performance remains acceptable. We believe as VLM capabilities continue to improve, this classifier-policy gap will naturally narrow, making the approach increasingly practical while maintaining bounded performance degradation.
>
> ### Q6: Generalizability of Difficulty Categories
>
> 1. Please check #1.2
>
>
> ## #3 Addressing the Limitations
>
> 1. Please check #1.1.
>
> 2. Please check #1.2.
>
> 3. Please check #1.3.
>
> 4. Please check #1.4.

---

> > ### Comment · Area_Chair_e2vk · 2025-08-04
> >
> > Dear Reviewer,
> >
> > The author response for paper "Dynamic Test-Time Compute Scaling in Control Policy: Difficulty-Aware Stochastic Interpolant Policy" is now available, and I kindly ask that you review it carefully and consider whether your concerns have been adequately addressed. If so, please acknowledge this in the discussion thread; if not, feel free to follow up with clarifying questions.
> >
> > Thank you for your contribution. AC

---

> > ### Comment · Reviewer_XJPo · 2025-08-04
> > **Rebuttal Received**
> >
> > Thank you for addressing the comments and providing thorough responses.

---

### Decision · Program_Chairs · 2025-09-17

**Decision:**

Accept (poster)

**Comment:**

The paper introduces the Difficulty-Aware Stochastic Interpolant Policy (DA-SIP), a test-time scaling approach for robotic control policies that adaptively allocates compute based on task difficulty using a difficulty classifier and the stochastic interpolant framework.

Strengths include a solid technical foundation built on the SI framework, addressing the inefficiency of fixed inference budgets in diffusion and flow-based robot policies, and demonstrating significant improvements in computation time without sacrificing task success rates.

Several weaknesses, such as reliance on simulation, annotation bottlenecks, and task-specific difficulty categories, were acknowledged by reviewers.
In rebuttal, the authors provided preliminary real-robot results (PushT) and additional experiments showing reduced data requirements and VLM-based generalization to unseen tasks. These responses were well received and alleviated many of the original concerns.

The paper received generally positive evaluations, with final scores of 5 (Accept) and 4, 4, 4 (Borderline Accepts). Given the novelty, technical soundness, strong empirical results, and the constructive rebuttal that improved reviewer confidence, I recommend Accept.